

# Quantifying biological carbon pump pathways with a data-constrained mechanistic model ensemble approach

Michael R. Stukel[1,2], Moira Décima[3], Michael R. Landry[3]

[1]Dept. of Earth, Ocean, and Atmospheric Science, Florida State University, Tallahassee, FL
[2]Center for Ocean-Atmospheric Prediction Studies, Florida State University, Tallahassee, FL
[3]Scripps Institution of Oceanography, University of California San Diego, San Diego, CA

*Correspondence to*: Michael R. Stukel (mstukel@fsu.edu)

**Abstract.** The ability to constrain the mechanisms that transport organic carbon into the deep ocean is complicated by the multiple physical, chemical, and ecological processes that intersect to create, transform, and transport particles in the ocean. In this manuscript we develop and parameterize a data-assimilative model of the multiple pathways of the biological carbon pump (NEMURO$_{BCP}$). The mechanistic model is designed to represent sinking particle flux, active transport by vertically migrating zooplankton, and passive transport by subduction and vertical

mixing, while also explicitly representing multiple biological and chemical properties measured directly in the field (including nutrients, phytoplankton and zooplankton taxa, carbon dioxide and oxygen, nitrogen isotopes, and $^{234}$Thorium). Using 30 different data types (including standing stock and rate measurements related to nutrients, phytoplankton, zooplankton, and non-living organic matter) from Lagrangian experiments conducted on 11 cruises from four ocean regions, we conduct an objective statistical parameterization of the model and generate one million

different potential parameter sets that are used for ensemble model simulations. The model simulates in situ parameters that were assimilated (net primary production and gravitational particle flux) and parameters that were withheld ($^{234}$Thorium and nitrogen isotopes) with reasonable accuracy. Model results show that gravitational flux of sinking particles and vertical mixing of organic matter from the surface ocean are more important biological pump pathways than active transport by vertically migrating zooplankton. However, these processes are regionally

variable, with sinking particles most important in oligotrophic areas of the Gulf of Mexico and California, sinking particles and vertical mixing roughly equivalent in productive regions of the CCE and the subtropical front in the Southern Ocean, and active transport an important contributor in the Eastern Tropical Pacific. We further find that mortality at depth is an important component of active transport when mesozooplankton biomasses are high, but that it is negligible in regions with low mesozooplankton biomass. Our results also highlight the high degree of

uncertainty, particularly amongst mesozooplankton functional groups, that is derived from uncertainty in model parameters, with important implications for results that rely on non-ensemble model outputs. We also discuss the implications of our results for other data assimilation approaches.



# 1. INTRODUCTION

Marine phytoplankton in the surface ocean are responsible for approximately half of the world's photosynthesis (Field et al., 1998). However, as a result of their short lifetimes and active grazing by a diverse suite of zooplankton, most of the carbon dioxide fixed by phytoplankton will be respired back into the surface ocean on a time scale of days to weeks (Steinberg and Landry, 2017). Long-term sequestration of this biologically-fixed carbon dioxide requires that the organic matter produced by marine autotrophs be transported into the deep ocean through a suite of processes collectively referred to as the biological carbon pump (BCP) (Boyd et al., 2019; Ducklow et al., 2001; Volk and Hoffert, 1985). The BCP is estimated to transport 5 – 13 Pg C $yr^{-1}$ into the deep ocean (Laws et al., 2011; Laws et al., 2000; Siegel et al., 2014; Henson et al., 2011). Our ability to constrain the magnitude of this globally important process (and its response to anthropogenic forcing) more accurately is hampered, however, by the diverse spatiotemporal scales over which these processes act and difficulties in quantifying rates in a heterogeneous three-dimensional ocean (Siegel et al., 2016; Burd et al., 2016; Boyd, 2015).

Attempts to predict future changes in the BCP are also complicated by the diverse pathways of organic matter flux into the deep ocean. Most research of the BCP has focused on sinking particles (Turner, 2015; Buesseler and Boyd, 2009; Martin et al., 1987; Honjo et al., 2008), which include diverse biologically-produced material such as dead phytoplankton and zooplankton, fecal pellets, crustacean molts, and mucous feeding structures (Smayda, 1970; Kirchner, 1995; Bruland and Silver, 1981; Fowler and Small, 1972; Small et al., 1979; Alldredge, 1976; Hansen et al., 1996; Lebrato et al., 2013). Slowly-sinking and suspended particles are also reshaped into rapidly-sinking marine snow through abiotic aggregation processes (Passow et al., 1994; Burd and Jackson, 2009; Jackson, 2001; Alldredge, 1998). These sinking particles are continually transformed, remineralized, and modified by a community of particle-attached bacteria and protists and suspension- and flux-feeding mesozooplankton (Stukel et al., 2019a; Poulsen and Kiorboe, 2005; Steinberg et al., 2008; Simon et al., 2002; Boeuf et al., 2019).

Organic matter is also transported into the deep ocean through active transport by vertically-migrating zooplankton and nekton (Steinberg et al., 2000; Longhurst et al., 1990; Archibald et al., 2019; Bianchi et al., 2013a) and by passive transport of dissolved and particulate organic matter that is subducted by ocean currents or mixed into the deep ocean (Levy et al., 2013; Carlson et al., 1994). The global magnitudes of these processes are highly uncertain because they are difficult to constrain from in situ measurements. Active transport is commonly believed to be responsible for a relatively small proportion (~10-20%) of the biological pump (Archibald et al., 2019; Hannides et al., 2009; Steinberg et al., 2000). However, if mortality at depth is included as part of active flux, it can be an important and at times dominant source of export, although such estimates are highly uncertain (Kelly et al., 2019; Kiko et al., 2020; Hernández-León et al., 2019). Similarly, investigations of the importance of passive transport initially focused on the role of refractory dissolved organic matter (Carlson et al., 1994; Copin-Montégut and Avril, 1993). Recent studies, however, highlight the importance and spatiotemporal variability of passive transport of particles via subduction, eddy mixing, mixed layer shoaling, and vertical diffusion (Levy et al., 2013; Omand et al., 2015; Stukel et al., 2018b; Stukel and Ducklow, 2017; Resplandy et al., 2019). These passive transport processes can be driven both by large-scale flows and by meso- and submesoscale circulation near fronts and eddies (Resplandy et al., 2019; Llort et al., 2018; Omand et al., 2015; Stukel et al., 2017).





70       Numerical models are essential tools for investigating such processes that act across multiple spatiotemporal scales and integrate multiple physical, chemical, and biological drivers. Such models have, for instance, been crucial in elucidating spatial and temporal decoupling of phytoplankton production and sinking particle export (Plattner et al., 2005; Henson et al., 2015), determining the temporal horizon over which anthropogenic signals appear in the world ocean (Schlunegger et al., 2019), quantifying regional variability in subduction of organic matter

(Levy et al., 2013), and predicting climate change impacts on plankton communities and the BCP (Dutkiewicz et al., 2013; Hauck et al., 2015; Bopp et al., 2005; Oschlies et al., 2008; Yamamoto et al., 2018). Models have also been used to investigate the role of vertically migrating zooplankton in strengthening oxygen minimum zones (Bianchi et al., 2013a), meso- and submesoscale hotspots of particle subduction (Resplandy et al., 2019), and the impact of glacial/interglacial changes in iron deposition on the BCP (Parekh et al., 2006). Such investigations would be

difficult or even impossible to undertake without models. Nevertheless, the models for varying processes differ substantially, and few are able to thoroughly investigate the full potential parameter space or quantify the accuracy of simulated energy flows through multiple trophic levels. While accurate simulation of physical circulation is critical for simulating marine biogeochemistry (Doney et al., 2004), objective parameterization of biogeochemical models lags substantially behind similar approaches for physics. Indeed, the inability to constrain biogeochemical

relationships accurately may be the primary limitation on our ability to objectively evaluate biogeochemical models (Anderson, 2005; Franks, 2009; Follows and Dutkiewicz, 2011; Ward et al., 2013). Recent advances in formal assimilation of biogeochemical properties into ocean models are beginning to allow objective model parameterization, a crucial first step for treating models as testable hypotheses (Xiao and Friedrichs, 2014a; Mattern and Edwards, 2019; Kaufman et al., 2018; Ford et al., 2018; Kriest et al., 2017; Shen et al., 2016; Oschlies, 2006).

Nevertheless, most of these approaches rely only on the assimilation of surface chlorophyll and/or other phytoplankton properties, thus leading to potentially high inaccuracies in parameterizing zooplankton dynamics (Shropshire et al., 2020; Löptien and Dietze, 2015). This is particularly important, because inaccurate parameterizations of mesozooplankton may lead to qualitatively and quantitatively inaccurate export dynamics (Cavan et al., 2017; Anderson et al., 2013). Accurate simulation of the BCP likely requires a focus on assimilation

of data types crossing multiple trophic levels and both ecological and biogeochemical parameters.

      In this study, we modify an existing, widely used biogeochemical model (NEMURO, Kishi et al., 2007) to focus specifically on the multiple pathways of the biological carbon pump. We refer to the new model as NEMURO$_{BCP}$. We have three distinct goals in creating NEMURO$_{BCP}$. The first is to mechanistically model the multiple BCP pathways (sinking particles, active transport by vertical migrants, and passive transport of organic

matter by ocean circulation and mixing). Our second goal is to enable direct comparability between model results and field measurements of standing stocks and rates. This allows the model to act as a synthetic tool using diverse measured variables to enhance investigation of underlying and unmeasured processes (Dietze et al., 2013). Our third goal is a model that can be run efficiently in multiple physical configurations to allow extensive data assimilation and hypothesis testing. NEMURO$_{BCP}$ is designed with a "core" nitrogen-based module (including all

biological components, nutrients, detritus, dissolved organic matter, and oxygen) that includes all three pathways of the BCP, along with submodules (that can be turned on or off) that model the carbon system, $^{234}$Th dynamics, and nitrogen isotopes. Here, we perform a Markov Chain Monte Carlo statistical data assimilation to develop ensemble parameterizations of NEMURO$_{BCP}$ using 30 distinct types of field measurements from 49 Lagrangian experiments.



We then investigate the model's ability to predict withheld measurements, conduct sensitivity analyses, and use the
model to investigate the BCP in four ocean regions.

## 2. METHODS

### 2.1. Core NEMURO$_{BCP}$ model

NEMURO$_{BCP}$ was developed from the NEMURO class of models originally developed for the North Pacific
(Kishi et al., 2011; Kishi et al., 2007; Yoshie et al., 2007) and includes several modifications adapted by Shropshire
et al. (2020) that allow the model to be compared more directly to common field measurements. It also includes
three optional modules that can be toggled on and off (the carbon system, nitrogen isotopes, and $^{234}$Th).

The core of NEMURO$_{BCP}$ is nitrogen-based and includes 19 state variables (Table 1): 3 nutrients (nitrate,
ammonium, and silicic acid), 2 phytoplankton (small phytoplankton and diatoms), 5 zooplankton (protistan
zooplankton, small non-vertically-migrating mesozooplankton, small vertically-migrating mesozooplankton, large
non-vertically-migrating mesozooplankton, large vertically-migrating mesozooplankton), 2 dissolved organic pools
(labile dissolved organic nitrogen and refractory dissolved organic nitrogen), 4 non-living particulate pools (small
particulate nitrogen, large particulate nitrogen, small opal, and large opal), two chlorophyll state variables (one
associated with small phytoplankton, the other with diatoms), and oxygen. As in Shropshire et al. (2020), the small
and large mesozooplankton are defined based on size (<1-mm and >1-mm, respectively) rather than trophic status to
allow direct comparison to common size-fractionated measurements. Relative to the original NEMURO model, key
changes include: 1) An explicit chlorophyll module (based on Li et al., 2010) that allows direct comparison to *in situ*
chlorophyll measurements and gut pigment measurements made with herbivorous zooplankton; 2) Division of
dissolved organic matter into refractory and labile dissolved organic nitrogen to simulate subduction of refractory
molecules; 3) Division of detrital pools into slowly and rapidly sinking particles to simulate more accurately the
gravitational pump; 4) Division of mesozooplankton into epipelagic resident taxa and vertical migrants to simulate
active transport by diel vertical migrators; and 5) Addition of a dissolved oxygen state variable that potentially limits
heterotrophic growth in the mesopelagic ocean. NEMURO$_{BCP}$ also modifies key transfer functions by, for instance,
allowing protists to feed on diatoms, since protistan grazers are often important diatom grazers (e.g., Landry et al.,
2011). The transfer functions linking state variables in NEMURO$_{BCP}$ are shown in Fig. 1 and explained in detail in
the online supplement. The 103 parameters in NEMURO$_{BCP}$ are explained in Supp. Table. S1.

Diel vertical migration is incorporated into NEMURO$_{BCP}$ via two alternate formulations. The first formulation
is designed for computational efficiency when the model is run in a euphotic zone only configuration
(NEMURO$_{BCP,EUPONLY}$). In NEMURO$_{BCP,EUPONLY}$ diel vertically migrating taxa (LZ$_{DVM}$ and PZ$_{DVM}$) only feed at
night. During the day, their mortality and respiration do not contribute to detritus and dissolved nutrient pools, but
rather are treated as a loss of nitrogen from the model. The second formulation includes a true diel vertical
migration model based on the model of Bianchi et al. (2013a) for use when the model explicitly represents
mesopelagic layers. In this formulation (NEMURO$_{BCP,DVM}$), vertically-migrating zooplankton swim towards a target
depth with a swimming speed of 3 cm s$^{-1}$ (with speed decreasing as zooplankton approach the target depth). During
the day, the target depth is defined as the depth of the $10^{-3}$ W m$^{-2}$ isolume. At night, the target depth is defined as



the mean depth of phytoplankton biomass. NEMURO$_{BCP,DVM}$ also includes a biological diffusion term that ensures
that LZ$_{DVM}$ and PZ$_{DVM}$ are dispersed around the target depth rather than accumulating within a single model layer.

### 2.1.1. Optional carbon system submodule

The carbon system in NEMURO$_{BCP}$ includes dissolved inorganic carbon (DIC) and alkalinity as state variables.
DIC is produced whenever there is net biological utilization of organic carbon and consumed whenever there is net
biological production of organic carbon at fixed stoichiometric ratios of C:N = 106:16 (mol:mol). Calculation of
other carbon system parameters (pH and partial pressure of $CO_2$) and air-sea $CO_2$ gas exchange are calculated using
procedures described in Follows et al. (2006).

### 2.1.2. Optional $^{234}$Th submodule

The $^{234}$Th submodule is based on the model of Resplandy et al. (2012). It adds a dissolved $^{234}$Th state variable,
as well as state variables associated with $^{234}$Th bound to each of the nitrogen-containing particulate state variables
(i.e., each phytoplankton, zooplankton, and detritus state variable). Dissolved $^{234}$Th is produced from the decay of
$^{238}$U (which is assumed to be proportional to salinity, Owens et al., 2011). Dissolved $^{234}$Th adsorbs onto the
aforementioned particulate pools following second-order rate kinetics. Particulate $^{234}$Th is returned to the dissolved
pool through both desorption and destruction of particulate matter. $^{234}$Th is also lost from the dissolved and
particulate phases through radioactive decay.

### 2.1.3. Optional $^{15}$N submodule

The nitrogen isotopes submodule is based on the NEMURO+$^{15}$N model of Stukel et al. (2018a) that was based
on an earlier isotope model by Yoshikawa et al. (2005). The $^{15}$N submodule adds an additional 13 state variables
that simulate the concentration of $^{15}$N in each nitrogen-containing state variable (nitrate, ammonium, all
phytoplankton and zooplankton groups, both detritus classes, and both dissolved organic nitrogen pools). Isotopic
fractionation occurs with most biological processes including nitrate uptake, ammonium uptake, exudation of
organic matter by phytoplankton, excretion and egestion by zooplankton, remineralization of detritus and dissolved
organic nitrogen, and nitrification.

### 2.2. Physical framework for model simulations

NEMURO$_{BCP}$ was developed so that it can be implemented in any physical framework. In this study, we used a
simple one-dimensional physical framework to simulate the water column associated with Lagrangian experiments
from which we derived our field data (see below). While this oversimplifies a system in which advection and
diffusion play important roles in re-distributing biological and chemical properties, we believe it is a reasonable
short-term approximation, especially because we are explicitly simulating results from *in situ* Lagrangian
experiments. In Lagrangian experiments, advection should play a reduced to negligible role in re-shaping plankton
time-series, although we note that Lagrangian drifters (see below) explicitly track only the mixed layer, which may
not be transported by the same currents as deeper layers. The use of a one-dimensional model does, however, allow
us to perform more than one million simulations for each of the 49 Lagrangian experiments, something that would
not be possible with a three-dimensional model grid. Our physical model framework simulates the euphotic zone





with variable vertical spacing that increases with depth, chosen to match sampling depths from the field programs. Vertical layers are centered at 2, 5, 8, 12, 20, 25, 30, 35, 40, 45, 50, 55, 60, 70, 80, 90, 100, 110, 120, 130, 140, 150, and 160 m, although for each Lagrangian experiment we only include depths above the 0.1% light level. We simulate vertical mixing as a simple diffusive process, with vertical eddy diffusivity coefficients varying with depth and estimated by Thorpe-scale analyses from field measurements (Gargett and Garner, 2008). Initial and boundary

conditions were determined from field measurements, although we sometimes had to estimate initial conditions from relationships with other measured parameters because all state variables were not measured for all experiments (e.g., if diatom biomass was not determined, we estimated it from a relationship between diatom biomass and total phytoplankton biomass). We ran the model for 30-days with constant vertical diffusion rates. 30-days is an arbitrary length of time to run the model, but this time span was chosen for multiple reasons: 1) it is long enough to

reduce sensitivity to initial conditions, 2) it is the longest period of time for which we would expect quasi-steady state conditions to be maintained in our study regions, 3) it allows sufficient time for parameter sets to potentially drive some taxa to near extinction (i.e., it allows time for unreasonable parameter sets to, for instance, lead to competitive dominance of small phytoplankton and drive diatoms to extinction). We recognize that maintaining constant physical forcing introduces inaccuracy to our simulations and hence expect model-data mismatches,

particularly during dynamic conditions (e.g., upwelling) when the system changes more rapidly. Model outputs were evaluated on the 30$^{th}$ day of the model simulation. Since we only simulate the euphotic zone, the model was run in NEMURO$_{BCP,EUPONLY}$ configuration.

### 2.3. Field data

Field data come from 49 short-term (~4-day) Lagrangian experiments conducted on 11 different cruises (Fig. 2)

in the California Current Ecosystem (CCE) (Ohman et al., 2013), in the Costa Rica Dome (CRD) in the Eastern Tropical Pacific (Landry et al., 2016a), in the Gulf of Mexico (GoM) (Gerard et al., in review), and at the Chatham Rise near the subtropical front in the Southern Ocean (Décima et al., in review). On these cruises a consistent sampling strategy involved utilization of an *in situ* incubation array with satellite-enabled surface drifter and 1×3-m "holey-sock" drogue centered at 15-m depth in the mixed layer (Landry et al., 2009). Samples for rate measurement

experiments (see below) were incubated in polycarbonate bottles placed in mesh bags at 6 – 8 depths spanning the euphotic zone on the incubation array (Landry et al., 2009). On 10 of the cruises, an identically-drogued sediment trap array was deployed to capture sinking particles (Stukel et al., 2015).

We assimilated a broad suite of standing stock and rate measurements across multiple trophic levels that included: 466 measurements of NO$_3^-$ concentration and 423 measurements of NH$_4^+$ concentration (Knapp et al.,

2021); 422 measurements each of silicic acid and 84 measurements of biogenic silica (Krause et al., 2016; Krause et al., 2015); 455 chlorophyll *a* measurements (Goericke, 2011); 193 measurements of small phytoplankton biomass by a combination of epifluorescence microscopy and flow cytometry (Taylor et al., 2012; Selph et al., 2021); 193 measurements of diatom biomass by epifluorescence microscopy (Taylor et al., 2012; Taylor et al., 2016); 193 measurements of protistan zooplankton biomass by epifluorescence microscopy and/or light microscopy of Lugol's

stained samples (Freibott et al., 2016); 44 measurements each of vertically-integrated <1- and >1-mm epipelagic-resident mesozooplankton biomass; 43 measurements each of vertically-integrated <1- and >1-mm diel-vertically-migrating mesozooplankton biomass; 413 measurements of particulate organic nitrogen and 28 measurements of dissolved organic nitrogen (Stephens et al., 2018); 342 measurements of net primary productivity by either H$^{13}$CO$_3^-$



or H$^{14}$CO$_3^-$ uptake methods (Morrow et al., 2018; Yingling et al., 2021); 149 measurements of nitrate uptake by
incorporation of $^{15}$NO$_3^-$ (Kranz et al., 2020; Stukel et al., 2016); 50 measurements of silicic acid uptake by
incorporation of $^{32}$Si (Krause et al., 2015); 248 measurements each of whole phytoplankton community growth rates
and whole phytoplankton community mortality rates due to protistan grazing determined by chlorophyll analyses of
microzooplankton dilution experiments (Landry et al., 2009; Landry et al., 2021); 53 measurements each of small
phytoplankton growth rates and small phytoplankton mortality rates due to protistan grazing determined by high-
pressure liquid chromatography pigment analyses of microzooplankton dilution experiments combined with flow
cytometry and epifluorescence microscopy (Landry et al., 2016b; Landry et al., 2021); 53 measurements each of
diatom growth rates and diatom mortality rates due to protistan grazing determined by high-pressure liquid
chromatography pigment analyses of microzooplankton dilution experiments combined with flow cytometry and
epifluorescence microscopy (Landry et al., 2016b; Landry et al., 2021); 41 measurements each of vertically-
integrated <1-mm and >1-mm nighttime mesozooplankton grazing rates by the gut pigment method (Décima et al.,
2016; Landry and Swalethorp, 2021); 41 measurements each of vertically-integrated <1-mm and >1-mm daytime
mesozooplankton grazing rates by the gut pigment method (Décima et al., 2016; Landry and Swalethorp, 2021); 37
measurements of sinking nitrogen using sediment traps (Stukel et al., 2019b; Stukel et al., 2021); 19 measurements
of sinking biogenic silica using sediment traps (Krause et al., 2016; Stukel et al., 2019b); and 475 measurements of
photosynthetically-active radiation. Each of the above measurements was typically the mean of measurements taken
at a specific depth (or vertically-integrated) on multiple days of the Lagrangian experiment, thus allowing us to also
quantify uncertainties for all measurements. Each of the above measurements also directly maps onto a specific
standing stock or process in the model enabling direct model-data comparisons. Field data are listed in Supp. Tables
S2 – S4.

**2.4. Data assimilation and objective model parameterization approach**

Using the available datasets described above, our goal was to develop an automated and objective model
parameterization method that would allow us to generate an ensemble of parameter sets for hypothesis testing or as
prior distributions in future data assimilation studies. We refer to this approach as objective ensemble
parameterization with Markov Chain Monte Carlo (OEP$_{MCMC}$). We began by log-transforming most field
measurements to normalize the data (some measurements, e.g. growth rates that can be positive or negative, were
not transformed). We then defined a cost function:

$$J(p) = \frac{1}{\sum \sqrt{N_{LE,i}}} \sum_{i=1}^{N_{sites}} \frac{\sqrt{N_{LE,i}}}{N_{DT,i}} \sum_{j=1}^{N_{DT,i}} \frac{1}{N_{O,i,j}} \sum_{k=1}^{N_{O,i,j}} \left(\frac{error_{i,j,k}}{unc_{i,j,k}}\right)^2$$

where $N_{sites}$ was the number of different sampling locations (i.e., 4 = CCE, CRD, GoM, and Chatham Rise), $N_{LE,i}$
was the number of Lagrangian experiments conducted at location $i$, $N_{DT,i}$ was the number of data types that were
measured at site $i$, $N_{M,i,j}$ was the number of distinct observations of data type $j$ at location $i$, and:

$$error_{i,j,k} = \begin{array}{ll} model_{i,j,k} - obs_{i,j,k} & \text{if } model_{i,j,k} > detlim_{i,j,k} \text{ or } obs_{i,j,k} > detlim_{i,j} \\ 0 & \text{if } model_{i,j,k} < detlim_{i,j,k} \text{ and } obs_{i,j,k} < detlim_{i,j} \end{array}$$



where $model_{i,j,k}$ is the model result corresponding to observation $obs_{i,j,k}$, and $detlim_{i,j,k}$ is the detection limit for data type $j$. This is equivalent to stating that there is no model data discrepancy if both the observation and the corresponding model result are below the experimental detection limit. Observational uncertainty was defined as:

$$unc_{i,j,k} = \max\left(\frac{\sigma_{i,j,k}}{\sqrt{N_{S,i,j,k}}}, ExpUnc_{i,j,k}\right)$$

where $\sigma_{i,j,k}$ is the standard deviation of multiple samples taken for the distinct observation $k$ of data type $j$ at location $i$ (i.e., $\sigma_{i,j,k}$ is the standard deviation of multiple measurements taken at the same depth over the course of a Lagrangian experiment), $N_{S,i,j,k}$ is the number of samples associated with observation $k$ of data type $j$ at location $i$, and $ExpUnc_{i,j,k}$ is the experimental uncertainty (e.g., instrument accuracy) of observation $k$ of data type $j$ at location $i$. We chose to use the maximum of these two terms because, in most cases, the standard error of repeated measurements was greater than experimental uncertainty (and inherently incorporates experimental uncertainty). However, in some cases (e.g., if three measurements of nitrate at 12 m depth on a particular Lagrangian experiment reported the same value), the standard error of the measurements was an unrealistically low estimate of true uncertainty.

The cost function, $J(p)$, gives equal weight to all measurement types within a specific Lagrangian experiment (e.g., if a Lagrangian experiment has 10 measurements of sinking nitrogen flux and 100 measurements of chlorophyll, $J(p)$ gives each of those measurement types equal weight). It also gives different locations a weight proportional to the square root of the number of Lagrangian experiments at that site. That decision was made so that a more heavily sampled region (i.e., CCE) can provide more constraint to the model, while preventing that region from overwhelming the model results. We note that this is a comparatively weak cost function (relative to, for instance, likelihood), because it normalizes to the number of measurements. We chose a weak cost function, because it reflects the fact that uncertainty in initial conditions and physical forcing introduces model data misfit that is unassociated with parameter choice.

To investigate the parameter space, we performed a Markov Chain Monte Carlo search (Metropolis et al., 1953). We first defined allowable ranges for all parameter values based on laboratory and field experiments, combined with results from prior model simulations (Supp. Table S1). These allowable ranges were defined to be broad and often spanned several orders of magnitude for a particular parameter. We then defined an initial guess for each parameter based primarily on values used in other NEMURO models (Kishi et al., 2007; Shropshire et al., 2020; Yoshie et al., 2007). We first ran 30-day simulations for all 49 Lagrangian experiments using the initial parameter values and calculated the cost function based on $J(p_1)$. We then perturbed the parameter set by adding to each parameter a random number drawn from a normal distribution with mean of 0 and standard deviation equal to a jump length of 0.02 times the width of the allowable range for that parameter. When newly selected values fell outside the allowable range, we mirrored them across the boundary. For many of the variables expected to follow a log-normal distribution (e.g., phytoplankton half-saturation constants), we log-transformed prior to the MCMC search. We then re-ran the 30-day model for all Lagrangian experiments and calculated a new cost associated with this parameter set, $J(p_2)$. We chose whether or not to accept this parameter set based on the relative cost functions of $J(p_1)$ and $J(p_2)$. If $J(p_2)$ was less than $J(p_1)$ we automatically accepted the new parameter set as a viable solution. If $J(p_2)$ was greater than $J(p_1)$, we accepted it with probability:



$$prob = e^{0.5 \times (J(p_n) - J(p_{n+1}))}$$

We walked through the parameter solution space for a total of 1.1 million iterations (discarding the first 100,000 iterations as a "burn-in" period before the cost function stabilized at a relatively low value). In this way, we explored the correlated uncertainty in all parameters of the core model, except the temperature sensitivity coefficient. We chose not to vary the temperature sensitivity coefficient (TLIM), because it is fairly well-constrained from experimental measurements and most model rates were directly correlated to TLIM; hence

changes in TLIM lead to commensurate changes in so many other rate parameters that allowing it to vary would have made calculation of mean values of other parameters (e.g., maximum growth or grazing rates) almost meaningless.

   We also saved model results associated with the BCP (e.g., sinking particle flux, net primary production, subduction rates, active transport) for the model simulations associated with each parameter set.

**3. RESULTS**

**3.1. Objective model parameterization**

   In our Markov Chain Monte Carlo (MCMC) exploration of the solution space, the cost function evaluated at our initial guess was 972. Over the first ~100,000 iterations of the MCMC procedure, the cost function declined to approximately 100 and remained near this value for the remainder of the MCMC procedure (1 million additional

simulations). We thus considered the first 100,000 iterations to be a "burn-in" period, and all results are based on the subsequent 1,000,000 solution sets. For this analysis set, the mean cost function was 98.2 with 95% confidence interval = 83.8 – 115.3. For comparison, we also conducted an undirected MCMC exploration of the solution space (i.e., every solution was accepted regardless of relative change in cost function) that yielded a mean cost function of 3197 (C.I. = 1270 – 5657) after the burn-in period, with a minimum value of 740 (across the 1,000,000 simulations).

The OEP$_{MCMC}$ procedure thus determined a set of 1,000,000 solutions for which the cost function was substantially reduced relative to either our initial parameter guess or a random sample of the solution space.

   We investigated the 1,000,000 OEP$_{MCMC}$ solution sets to determine which parameters were well or poorly constrained by the data (Supp. Tables S1 and S2). We focus here on how well the field observations allowed the OEP$_{MCMC}$ approach to constrain the parameters relative to prior estimates of allowable ranges. This is distinct from

the question of which parameters are most well constrained because some parameters were well known from prior knowledge (e.g., phytoplankton maximum growth rates) while others are poorly known (e.g., phytoplankton half-saturation constants). Some parameters were very well constrained by the data. Ten of the 101 variables were constrained to within 10% of their allowed range (for log-transformed variables, 10% of their log-transformed parameter space). Six of the 10 well-constrained variables were associated with phytoplankton bottom-up forcing,

while only two parameters associated with zooplankton were well constrained by the data (the Ivlev constants for protistan grazing on small and large phytoplankton). The most well-constrained parameter was the ammonium half-saturation constant for small phytoplankton which was assumed to vary from 0.001 – 1 mmol NH$_4^+$ m$^{-3}$ and was constrained by the OEP$_{MCMC}$ procedure to a 95% C.I. of 0.0011 - 0.0065 mmol NH$_4^+$ m$^{-3}$. For metazoan zooplankton, all parameters except Ivlev constants had 95% C.I.s that spanned >60% of the allowable range, and

many exceeded 90% of the allowable range. Overall, 25 parameters had 95% C.I.s that spanned >60% of the





allowable range, suggesting that those parameters were more strongly constrained by our prior estimates than by the field data (Supp. Table S1).

Next, we highlight analyses of bottom-up forcing on small phytoplankton (Fig. 3) and correlation of large phytoplankton (i.e., diatoms) bottom-up forcing with other model dynamics (Fig. 4) as examples of typical patterns

of correlation among parameters. Small phytoplankton parameters were generally well-constrained by the extensive datasets of phytoplankton growth rates, net primary production, and phytoplankton biomass (as assessed microscopically and/or by chlorophyll analyses). For instance, although we allowed the maximum growth rate of small phytoplankton ($V_{max,SP}$) to vary from 0.1 to 1 $d^{-1}$, the OEP$_{MCMC}$ procedure constrained $V_{max,SP}$ to 0.22 to 0.64 (at the 95% C.I.). The least well constrained parameter related to small phytoplankton growth was the half-

saturation constant for nitrate uptake, which varied from 0.011 to 1.3 mmol N $m^{-3}$. Several of these phytoplankton parameters were also correlated in predictable manners. For instance, $V_{max,SP}$ was negatively correlated with the initial-slope of the photosynthesis-irradiance curve ($\alpha_{SP}$, correlation coefficient ($\rho$) = -0.15), because increased maximum growth rates and stronger light dependence (i.e., a slower rate of increase in photosynthesis with increasing light) offset each other to maintain similar realized growth rates under typical light-limited conditions.

$V_{max,SP}$ was also positively correlated with the mortality rate (mort$_{SP}$, $\rho$=0.25), because commensurate changes in $V_{max,SP}$ and mort$_{SP}$ maintain similar net growth rates for small phytoplankton.

Parameters associated with large phytoplankton were typically less well-constrained, although they did differ from parameters associated with small phytoplankton in several predictable ways. For instance, the maximum growth rate of large phytoplankton ($V_{max,LP}$, mean = 0.72 $d^{-1}$, 95% C.I. was 0.43 – 0.99 $d^{-1}$) was greater than the

maximum growth rate of small phytoplankton (mean = 0.37 $d^{-1}$, 95% C.I. was 0.22 – 0.64 $d^{-1}$) despite the fact that we used identical allowable ranges of 0.1 – 1 $d^{-1}$. The half-saturation rate for large phytoplankton uptake of nitrate ($K_{NO,LP}$ = 1.6 mmol N $m^{-3}$) was also substantially greater than $K_{NO,SP}$ (0.25 mmol N $m^{-3}$), although their half-saturation constants for ammonium uptake were similar. Unsurprisingly, the maximum growth rate of large phytoplankton was strongly correlated with the maximum growth rate of protistan zooplankton on large

phytoplankton ($g_{max,SZ,LP}$, $\rho$=0.35), because grazing by protistan zooplankton is often the dominant source of mortality for all phytoplankton (including diatoms). More surprisingly, $V_{max,LP}$ had an even stronger correlation with the maximum grazing rate of epipelagic-resident large (>1-mm) mesozooplankton on small phytoplankton ($g_{max,PZRES,SP}$, $\rho$ = 0.43). We believe that this arises from a correlation between large mesozooplankton standing stock and $g_{max,PZRES,SP}$. Since small phytoplankton are often the most abundant potential prey item, higher

$g_{max,PZRES,SP}$ values allow large mesozooplankton (which preferentially graze large phytoplankton) to sustain higher biomass and prevent large phytoplankton from escaping grazing pressure, thus requiring a higher maximum growth rate to maintain their biomass.

Some correlations were unexpected. For instance, the initial slope of the photosynthesis-irradiance curve ($\alpha_{LP}$) was positively correlated with the remineralization rate of labile dissolved organic nitrogen to $NH_4^+$ (ref$_{dec,DON,NH}$,

$\rho$=0.31). Both of these parameters were strongly constrained by the OEP$_{MCMC}$ procedure ($\alpha_{LP}$ had an allowable prior range of 0.001 – 0.04 $m^2$ $W^{-1}$ $d^{-1}$ but had a posterior 95% C.I. of 0.008 – 0.03 $m^2$ $W^{-1}$ $d^{-1}$; ref$_{dec,DON,NH}$ had an allowable range of 0.005 – 0.3 $d^{-1}$ but a 95% C.I. of 0.005 – 0.01 $d^{-1}$). It is not clear why these parameters would be correlated, although it is likely related to the relative realized growth rates of large phytoplankton in the upper and lower euphotic zone. High values of $\alpha_{LP}$ would promote higher realized growth rates in the deep euphotic zone;



high values of ref$_{dec,DON,NH}$ would lead to higher realized growth rates in the nutrient-limited upper euphotic zone. The Ikeda parameter for small mesozooplankton (Ik$_{LZ}$, d$^{-1}$), which sets the basal respiration of small (<1-mm) mesozooplankton was positively correlated with V$_{max,LP}$ ($\rho = 0.12$), K$_{NH,LP}$ ($\rho = 0.16$), and $\alpha_{LP}$ ($\rho = 0.29$). While the first and third correlations are not surprising (both lead to increased large phytoplankton growth which would support higher mesozooplankton respiration), it is surprising that Ik$_{LZ}$ would be correlated with K$_{NH,LP}$ since higher

half-saturation constants lead to lower realized phytoplankton growth rates. V$_{max,LP}$ was also negatively correlated with the daytime mortality rate of small (<1-mm) vertically-migrating mesozooplankton at their mesopelagic resting depth (mort$_{day,LZDVM}$, $\rho = -0.35$), which is opposite to what would be expected if large phytoplankton growth was necessary to support mesozooplankton mortality, but may reflect an indirect effect of intraguild competition between small mesozooplankton and protistan grazers (mort$_{day,LZDVM}$ was also negatively correlated with the Ivlev constant

for small mesozooplankton grazing on protistan zooplankton (Iv$_{LZDVM,SZ}$, $\rho = -0.27$) which would indicate that mesozooplankton increases when their feeding rate on protists increases).

        While these are only a subset of the multiple correlations, they highlight the complex, and often counterintuitive, relationships among many parameters. This analysis also clearly elucidates the importance of joint parameter sensitivity analyses. For instance, when model sensitivity to maximum large vertically-migrating

mesozooplankton grazing rates on small phytoplankton (g$_{max,PZRES,SP}$) was investigated with a maximum large phytoplankton growth rate (V$_{max,LP}$) of ~0.6 d$^{-1}$, the analysis suggested that the model was only weakly sensitive to g$_{maxPZRES,SP}$, and that the optimal value was near 0.03 d$^{-1}$. However, when the same analysis was conducted with V$_{max,LP}$ = ~1.0, the model was very sensitive to g$_{maxPZRES,SP}$, and the optimal value was 0.1 – 0.2 d$^{-1}$.

### 3.2. Model data comparison (assimilated data)

To determine whether the model was able to simulate assimilated measurements accurately, we compared model-data results with respect to two key processes related to export: net primary production and sinking particle flux (Figs. 5 and 6, respectively). For most Lagrangian experiments, the model 95% confidence interval bracketed the mean of the observed net primary production (Fig. 5). However, the model substantially underestimated net primary productivity for several experiments in the CCE (e.g., 605-1, 605-3, 704-4, 810-5, and 1604-4) conducted in

near-coastal regions with recently upwelled high-nitrate water. The model-data discrepancy thus likely results from our assumption of a one-dimensional system with constant physics for 30-days. In reality, these Lagrangian experiments were heavily influenced by coastal upwelling processes missing in our one-dimensional model and experienced markedly non-linear dynamics as the water parcels were advected away from the upwelling source and nutrients drawn down over time (e.g., Landry et al., 2009). Contemporaneous nutrient input from directly below

these water parcels was thus likely not the source of nitrogen supporting high production, as is assumed by our one-dimensional physical framework. In less dynamic regions (e.g., GoM), the model more faithfully simulated phytoplankton production.

        The model did a reasonable job simulating sinking particle export flux from the euphotic zone (Fig. 6). For the majority of experiments, observed export fell within the 95% confidence interval of the model simulations.

However, the simulated export flux range was quite substantial for most cycles. Indeed, the 95% confidence intervals for export flux at single locations using the 1,000,000 MCMC solutions were at times as large as the confidence interval for mean observed flux across the 49 different Lagrangian experiments. This suggests that



uncertainty in parameter estimation for the model is as important a source of error for export flux as variability

between regions and seasons. The only region for which the model produced a stark bias in export flux relative to

the observations was the CRD, where the model consistently overestimated export flux. This is not surprising for

this region, because the CRD is dominated by *Synechococcus*, which contribute substantially less to export flux than

larger phytoplankton (Saito et al., 2005; Stukel et al., 2013). In other regions, model underestimates of export flux

were typically more notable than model overestimates (observations were seldom less than the lower bound of the

model's 95% confidence interval).

**3.3. Model data comparison (unassimilated data)**

To assess the model's ability to simulate state variables and processes not included in the assimilation dataset,

we utilized the thorium sorption and nitrogen isotope submodules and compared model results to measured total

water column $^{234}$Th (Fig. 7), the C:$^{234}$Th ratio of sinking particles (Fig. 8a), and the $\delta^{15}$N of sinking particles (Fig.

8b). NEMURO$_{BCP}$ accurately simulated many properties of $^{234}$Th dynamics found in the field data. For instance, it

did a reasonable job of estimating the shape and magnitude of vertical profiles, notably simulating low $^{234}$Th activity

in surface waters and $^{234}$Th activity close to equilibrium with $^{238}$U in deeper waters. The model also captured some

key aspects of inter- and intra-regional variability in $^{234}$Th activity, including much lower $^{234}$Th activity in coastal

regions of the CCE (e.g., Fig. 7a, c, ah) relative to offshore regions (e.g., Fig. 7e, ad, ae). The model also accurately

estimated the consistently high $^{234}$Th activity found in the GoM. The greatest model-data mismatch with respect to

$^{234}$Th activity was found in the CRD (Fig. 7ai – am). In this region, the model was fairly accurate at predicting

mixed layer $^{234}$Th activity, but the model consistently underestimated $^{234}$Th activity in the deep euphotic zone. The

model was also reasonably effective at predicting the C:$^{234}$Th ratio of sinking particles. The model both accurately

estimated the mean value of sinking particle C:$^{234}$Th ratios (median observation = 7.2 μmol C dpm$^{-1}$; median model

value for locations paired with observations = 7.7 μmol C dpm$^{-1}$) and the range of C:$^{234}$Th values (observation = 2.2

– 20.5 μmol C dpm$^{-1}$; model = 4.1 – 30.0 μmol C dpm$^{-1}$). For most simulations, the modeled and observed C:$^{234}$Th

ratios also showed very good agreement (Fig. 8a). However, the model consistently overestimated the C:$^{234}$Th ratio

of sinking particles in the CRD, a region where the model was particularly poorly constrained and predicted a wide

range of C:$^{234}$Th ratios. The model also substantially underestimated the C:$^{234}$Th ratio for several sediment trap

collections in the GoM. Nevertheless, the overall model-data agreement with respect to $^{234}$Th dynamics is

reassuring, especially since key parameters (e.g., thorium sorption and desorption coefficients) were not estimated

by the OEP$_{MCMC}$ procedure but instead were taken directly from the literature.

The model was also able to accurately simulate the $\delta^{15}$N of sinking particles, albeit with a more limited set of

observations available (note that we did not simulate nitrogen isotopes for Lagrangian experiments from the

SalpPOOP cruise, because the $\delta^{15}$N of deepwater nitrate, an important boundary value, was unknown in this region).

The median observed $\delta^{15}$N of sinking particles was 4.6 compared to a model estimate of 6.1, while the observed

range was 1.7 – 14.3 and the modeled range was 1.8 – 9.3 (Fig. 8b). The only simulation for which there was a

substantial mismatch between model result and observation was from a single experiment in the CRD in which there

is substantial uncertainty in the observed $\delta^{15}$N because one sediment trap replicate had a very high $\delta^{15}$N value, while

the other two replicates had values near the simulated value.

**3.4. Sensitivity analysis**



The OEP$_{MCMC}$ approach allowed us to investigate uncertainty associated with all three pathways of the BCP (see the next two sections). First, we focus specifically on variability in model estimates of gravitational flux, as these can be directly compared to field measurements. When comparing modeled gravitational flux for different Lagrangian cycles, the median coefficient of variation (standard deviation / mean) was 0.49, with a range of 0.29 –
1.38. This represents substantial uncertainty in sinking particle flux due solely to different potential parameter choices (Fig. 6). For instance, on the fifth Chatham Rise Lagrangian experiment (which was the experiment with coefficient of variation closest to the median), the mean model predicted gravitational flux was 1.24 mmol N m$^{-2}$ d$^{-1}$ with a standard deviation of 0.62 mmol N m$^{-2}$ d$^{-1}$ and a 95% confidence interval from 0.29 to 2.6 mmol N m$^{-2}$ d$^{-1}$. This shows that for a typical cycle, there was nearly an order of magnitude variability in export flux based solely on
uncertainty in model parameterization. For comparison, across the 49 Lagrangian experiments for which we have sediment trap deployments near the base of the euphotic zone, the field observations of gravitational flux at the base of the euphotic zone ranged from 0.22 – 6.3 mmol N m$^{-2}$ d$^{-1}$. Thus, for a typical Lagrangian experiment, uncertainty in model parameterization introduced slightly less uncertainty in gravitational flux than variability across the multiple regions. For the fourth GoM Lagrangian experiment (the experiment with the highest coefficient of
variation), the mean model predicted gravitational flux was 0.23 mmol N m$^{-2}$ d$^{-1}$ with a standard deviation of 0.31 and a 95% confidence interval from 0.0069 – 1.07 mmol N m$^{-2}$ d$^{-1}$. For this particular cycle, some likely parameter sets predicted gravitational flux nearly equal to the mean measured gravitational flux across the diverse regions we studied, while other likely parameter sets predicted export more than an order of magnitude lower than the lowest observed flux. This high degree of uncertainty should be considered when results of a single model simulation are
considered and provide a strong argument for the importance of ensemble modeling.

To investigate the relationships among uncertainties in the three pathways of the BCP and uncertainties in parameters, we computed the $R^2$ of ordinary least squares linear regressions of each BCP pathway as a function of each parameter. This approach allows us to quantify the percentage of variability in the export pathway explained by a linear relationship with a specific parameter. This is distinctly different from some traditional sensitivity
analysis approaches that either compute the derivative of a model output with respect to different parameters or vary parameters by a fixed amount (e.g., ±10%). Unlike those approaches, our $R^2$ approach explicitly accounts for the certainty with which different parameters are constrained. For instance, a model may be very sensitive to the maximum growth rate of diatoms; however, if that parameter is well constrained by laboratory experiments, field data, and/or data assimilation, then parameter uncertainty may not be the dominant source of uncertainty in model
results. Our approach is thus well suited to determining which parameters especially merit future experimental focus.

Our results show that the $R^2$ values for BCP pathways regressed against most parameters were ~0.01 or less. However, some of the parameters were able to explain 10% of the variability in specific BCP pathways. For instance, the linear mortality parameter for protistan zooplankton (mort$_{SZ}$) explained 15% of the variability in
gravitational particle export (positive correlation) and 18% of the variability in export due to vertical mixing (negative correlation). These correlations reflect the importance of protistan zooplankton in controlling phytoplankton populations without producing rapidly sinking particles. Multiple parameters had similar inverse correlations with gravitational particle export and export due to vertical mixing. For example, the assimilation efficiency of small epipelagic-resident mesozooplankton, the Ivlev constant for large mesozooplankton feeding on
small mesozooplankton, and the sinking speed of fast-sinking detritus all had positive correlations with gravitational





flux; the maximum grazing rate of small epipelagic-resident mesozooplankton on protistan zooplankton, and the remineralization rate of fast-sinking detritus had negative correlations with gravitational flux. The remineralization rate of fast-sinking detritus explained the highest proportion of variability in gravitational flux (45%). Only two parameters (the maximum grazing rate of large vertically migrating mesozooplankton on small mesozooplankton

and the Ivlev constant for large mesozooplankton feeding on small protists) explained >10% of the variability in active transport (19% and 18%, respectively, with both positively correlated with active transport). Notably, none of the parameters most responsible for uncertainty in the BCP were related to phytoplankton bottom-up limitation. We do not believe that this reflects a lack of importance of bottom-up processes in the BCP. Rather, this reflects a much greater uncertainty in parameterizations for zooplankton and non-living organic matter, combined with the

importance of these processes to the BCP (Cavan et al., 2017; Anderson et al., 2013).

As mentioned previously, two of the most important parameters for determining gravitational flux are the sinking speed (Lsink) and remineralization rate of fast-sinking particles to DON ($ref_{dec,LPON,DON}$). Notably, these two parameters are strongly related to the remineralization length scale for these particles ($RLS = Lsink/(ref_{dec,LPON,DON} + ref_{dec,LPON,NH4})$). We illustrate the impact of variability in RLS on model gravitational

flux by focusing on two Lagrangian experiments representative of the CRD (CRD-1) and upwelling-influenced regions of the CCE (1604-3). RLS was strongly correlated with gravitational flux for each experiment (Pearson's $\rho$ = 0.62 for both experiments, $p \ll 10^{-7}$). The relationship was not perfectly linear, however (Supp. Fig. S1a,b). Particularly for the CRD experiment, but also for the CCE experiment, there was a threshold effect such that RLS was only weakly correlated with gravitational flux at RLS > ~150 m. This resulted from higher RLS values leading

to decreased recycling in the system and hence reduced primary production. Comparison of the probability density functions for RLS determined by the $OEP_{MCMC}$ procedure with probability density functions for only those parameter sets that accurately predicted gravitational flux for these cycles (to ±1 standard deviation of the observed value) show that gravitational flux was more accurately predicted for the CCE experiment with RLS values slightly higher than the overall average of the whole dataset (median for the entire dataset was 85 m; median for parameter

sets that accurately predicted export for this cycle was 115 m, Supp. Fig. S1c), while it was more accurately predicted for the CRD experiment with RLS values lower than the average for the dataset (median RLS for accurate parameter sets = 57 m, Supp. Fig. S1d). This highlights the sensitivity of the model to these parameters while suggesting differences in remineralization length scale between these specific regions, although we caution that RLS calculated above is only for fast-sinking detritus and does not account for the additional gravitational flux mediated

by slowly sinking particles.

### 3.5. Model results: Three pathways of export

We compared the relative magnitude of the three BCP pathways for all Lagrangian cycles and all $OEP_{MCMC}$ parameter sets (Fig. 9a). Results showed that export was typically dominated by some combination of gravitational and/or mixing flux. Active transport typically contributed a relatively small proportion of export from the base of

the euphotic zone (mean = 2.8%, 95% C.I. = 0.02% - 16.5%). Across the dataset, gravitational flux was the dominant export pathway (mean = 56.1%, 7.1% - 99.6%), although mixing was also an important source of export (mean = 41.1%, 0% - 92.3%). The large confidence intervals for each of these fluxes highlight the uncertainty in our estimates of the BCP pathways. They also, however, obscure distinct regional variability among the experiments analyzed in our study.



During upwelling-influenced experiments in the CCE, mixing and gravitational flux often contributed approximately equally to the BCP, with different parameter sets suggesting either dominance by mixing or gravitational flux. For instance, during CCE cycle 1604-3 (Fig. 9b) gravitational flux contributed an average of 61% (29 – 84%) of export, while mixing was responsible for 35% (12 – 67%). Not every CCE coastal cycle had a relatively even split, however, with some more dominated by sinking flux and others more dominated by mixing flux (e,g. CCE cycle 0605-3 which occurred during a dense coastal dinoflagellate bloom, Fig. 9g). In oligotrophic regions of the CCE and GoM, export was typically dominated by sinking flux, with relatively minor contributions from both mixing and active transport. For instance, during CCE cycle 1408-5 gravitational flux was responsible for 86% (70 – 97%) of export (Fig. 9c), while during GoM cycle 5 sinking was responsible for 89% (66 – 98%) of export (Fig. 9e). During CRD experiments, which had relatively high mesozooplankton biomasses relative to phytoplankton biomass, active transport was comparatively more important. For instance, during CRD cycle 1, active transport averaged 6.5% (0.7 – 26%) of export and was more important than mixing flux (4.3%, 0.4 – 12%, Fig. 9d). During the Chatham Rise experiments in the Southern Ocean, export patterns were comparable to those in the upwelling-influenced CCE, driven primarily by gravitational flux and mixing, with gravitational flux slightly more important.

Looking at patterns across regions and across the varying conditions on our Lagrangian experiments, the proportion of export driven by vertical mixing was correlated with vertical eddy diffusivity at the base of the euphotic zone (Spearman's $\rho = 0.64$, $p<10^{-6}$). This is not surprising, since vertical diffusion drives particulate and dissolved organic matter flux across the euphotic zone. Because sinking and vertical mixing were the two dominant mechanisms of export, vertical eddy diffusivity also showed a strong inverse correlation with gravitational flux (Spearman's $\rho = -0.64$, $p<10^{-6}$). Across all simulations, organic matter mixed out of the euphotic zone was relatively evenly split between DOM and POM, but variability in POM flux was greater (mean = $3.4 \pm 6.9$ mmol N $m^{-2}$ $d^{-1}$) than variability in DOM (mean = $4.6 \pm 5.5$ mmol N $m^{-2}$ $d^{-1}$). For most simulations (72%), DOM mixing flux exceeded POM mixing flux. However, POM mixing was greater for 66% of the simulations with total mixing flux >20 mmol N $m^{-2}$ $d^{-1}$. Flux of fast-sinking particles exceeded that of slow-sinking particles at the euphotic zone base for 90.5% of simulations, with fast-sinking particles averaging of 2.3 mmol N $m^{-2}$ $d^{-1}$ (0.07 – 10.4 mmol N $m^{-2}$ $d^{-1}$) and slow-sinking particles averaging 0.35 mmol N $m^{-2}$ $d^{-1}$ (0.02 – 1.4 mmol N $m^{-2}$ $d^{-1}$).

### 3.6. Model results: Diel vertical migration and active transport

In NEMURO$_{BCP}$, active transport is driven by two processes: respiration/excretion and mortality at depth. The former is parameterized as a temperature- and size-dependent function representing basal respiration and is comparatively well constrained by prior experimental work. The latter is parameterized as a density-dependent function representing predator-induced mortality, a process that is highly uncertain because few studies have quantified zooplankton mortality in the mesopelagic ocean. We fit linear regressions to log-transformed active transport plotted against log-transformed mesozooplankton biomass (Fig. 10a) to determine a power law relationship predicting active transport from mesozooplankton biomass: $AT = aB^c$, where AT = active transport (mmol N $m^{-2}$ $d^{-1}$), B = biomass (mmol N $m^{-2}$), a = $0.0052 \pm 6\times10^{-6}$, and c = $1.29 \pm 0.0004$, $R^2 = 0.90$, $p<<10^{-9}$. Similar relationships were also determined for the respiration/excretion component of active transport (E = $aB^c$, a = $0.0037 \pm 4\times10^{-6}$, b = $1.02 \pm 0.0005$, $R^2 = 0.87$, $p<<10^{-9}$) and the mortality component of active transport (M = $aB^c$, a = $0.00054 \pm 10^{-6}$, b = $2.04 \pm 0.001$, $R^2 = 0.89$, $p<<10^{-9}$). As expected, since excretion is density-independent while





mortality is density-dependent, the exponent of the excretion power law was ~1 and the exponent of the mortality

power law was ~2. This led to mortality becoming a greater fraction of total active transport as mesozooplankton

biomass increased (Fig. 10d). The transition from active transport dominated almost entirely by respiration to active

transport comprised mostly of mortality at depth occurred rapidly as biomass increased past ~5 mmol N m$^{-2}$. As a

result of the density-dependent parameterization of mortality, daytime mortality also increased with increasing

zooplankton biomass (m = aB$^c$, where m is specific mortality (h$^{-1}$) a = $2.6 \times 10^{-4} \pm 5 \times 10^{-6}$, and b = $0.995 \pm 0.001$, R$^2$ =

0.68, p<<10$^{-9}$). This generated daily mortality rates (i.e., over a 12-h daytime period) of ~0.3% d$^{-1}$ at a biomass of 1

mmol N m$^{-2}$ and ~6% d$^{-1}$ at a biomass of 20 mmol N m$^{-2}$ (Fig. 10e). Overall mortality for vertically-migrating

mesozooplankton was approximately evenly split between the epi- and mesopelagic, although this ratio was poorly

constrained by the model (Fig. 10f). For instance, 9% - 96% of large-mesozooplankton mortality occurred in the

mesopelagic (at the 95% C.I.).

As suggested by the validation data, vertical migrator biomass was primarily found in the large (>1-mm)

mesozooplankton size class. The large mesozooplankton were also predominantly vertical migrators, while the

small mesozooplankton were primarily epipelagic residents (Fig 10g). Consequently, large mesozooplankton

typically dominated active transport (Fig. 10h) even though small mesozooplankton usually contributed

proportionally more to active transport than to biomass as a result of higher specific respiration rates (Fig. 10i).

It would be reasonable to assume that predator-induced mortality in the deep ocean would be negatively

correlated with the abundance of diel-vertical migrators, because high mortality would yield a competitive

advantage for epipelagic-resident zooplankton. For the full dataset, however, we found a negligible correlation

between the mesopelagic mortality term for large mesozooplankton (mort$_{day,PZDVM}$) and large mesozooplankton

biomass (Spearman's ρ = -0.0077). When investigating this correlation for individual experiments, the correlation

was sometimes positive and sometimes negative. This lack of a correlation was driven by strong correlations

between the mort$_{day,PZDVM}$ and both the assimilation efficiency of these zooplankton and their maximum grazing rate

on smaller mesozooplankton. This led to a compensatory higher growth rate to offset the higher mortality rate and

consequently to a reasonably strong correlation between mort$_{day,PZDVM}$ and the magnitude of export attributable to

predation on large mesozooplankton in the deep ocean (ρ = 0.25).

**4.  DISCUSSION**

**4.1. Biological carbon pump pathways**

Gravitational flux is by far the most well studied pathway of the BCP, because it is the only pathway for which

direct *in situ* flux measurements are possible. Nevertheless, incredibly sparse *in situ* sampling necessitates

spatiotemporal extrapolation approaches to derive regional and global estimates of gravitational flux, including the

use of forward models, inverse models, and satellite algorithms (e.g., Schlitzer, 2004; Laws et al., 2000; Hauck et

al., 2015). Satellite algorithms, as perhaps the most widely used and cited methods for deriving global estimates,

deserve special attention. These approaches have delivered widely varying estimates of the magnitude of

gravitational flux, and indeed the algorithms underlying such estimates often differ in the fundamental relationship

predicted between sinking particle flux and phytoplankton biomass and production (Laws et al., 2000; Siegel et al.,

2014; Henson et al., 2011; Dunne et al., 2005). Such studies typically estimate export flux from relationships with

net primary production (or surface chlorophyll) and/or temperature because these properties are easily observable by



satellite remote sensing. These studies, however, have reached widely differing relationships about the relationships of these properties to export efficiency (*e*-ratio = gravitational flux / net primary productivity). Indeed, the *in situ* data compiled here shows no significant dependence of export efficiency on NPP or temperature (Figure 11a), because export efficiency depends not just on temperature and phytoplankton production, but also the community composition of phytoplankton and zooplankton, physiological adaptations of important taxa, and a multitude of ecological interactions (Turner, 2015; Buesseler and Boyd, 2009; Guidi et al., 2016). Indeed, focusing only on the regions studied here, anomalously high *Synechococcus* abundances likely result in low export efficiency in the CRD (Stukel et al., 2013; Saito et al., 2005), salp blooms drive very high export in the Chatham Rise (Décima et al., in review), and the diatom *Thalassiosira* seems to play a particularly important role in export in the CCE (Preston et al., 2019; Valencia et al., 2021). In the latter, diatom photophysiological health is a strong predictor of export (Brzezinski et al., 2015), although the diatoms likely sink mainly after grazing by metazooplankton (Morrow et al., 2018).

Despite the diversity of processes that affect the BCP, many of which are not included in NEMURO$_{BCP}$, our simulations reasonably reproduce the variability of export efficiency across the study regions, even though results for individual experiments are imprecise (Fig. 11). One important process that drives variability in export efficiency is temporal decoupling of production and export (Henson et al., 2015; Laws and Maiti, 2019; Kahru et al., 2020). Despite the use of constant physical forcing throughout our 30-day simulations, they exhibit distinct temporal variability in biogeochemical properties. We highlight results from one experiment in slightly aged, upwelled water off the California coast, using 5 different evenly spaced parameter sets chosen from our ensemble (Fig. 12). In each of these simulations, net primary production increases early in the simulations, rapidly in some, more gradual in others (Fig. 12a). Net primary production soon diverges in all of the simulations, however, with some gradually decreasing after the first week and others exhibiting blooms. Gravitational flux was even more variable, with one simulation peaking almost immediately and others with substantial temporal lags between net primary production and export (Fig. 12b). This led to substantial temporal variability in export efficiency (Fig. 12c) and quite complex relationships between gravitational flux and net primary production (Fig. 12d).

Assessing the accuracy with which the model simulates export due to vertical mixing (variously called the eddy subduction pump, mixed layer pump, and/or physical pump) is more difficult. Previous studies to quantify this process have either relied on indirect biogeochemical proxies (Stukel and Ducklow, 2017; Llort et al., 2018) or numerical models (Omand et al., 2015; Levy et al., 2013; Stukel et al., 2018b) to quantify these processes. Our vertical mixing results should be considered with some caution due to our overly simplified one-dimensional physical framework. Nevertheless, it is reassuring that our simulations from the CCE, which showed that vertical mixing out of the euphotic zone was often similar in magnitude to gravitational flux and at times even higher, is similar to results based on a Lagrangian particle model developed for the region (Stukel et al., 2018b). More realistic estimates for all regions could be derived by coupling NEMURO$_{BCP}$ and our parameter ensembles to a three-dimensional ocean simulation.

The magnitude of active transport mediated by diel-vertically migrating zooplankton in the global ocean remains highly uncertain due to a paucity of measurements and the difficulty of constraining the amount of mortality occurring at depth. Studies that include only respiration and/or excretion of zooplankton at depth typically find that active transport is a relatively small fraction of gravitational flux (Steinberg et al., 2000; Hannides et al., 2009).



However, more recent studies that have attempted to incorporate mortality experienced in the deep ocean have derived much larger estimates of active transport (Kelly et al., 2019; Kiko et al., 2020; Hernández-León et al., 2019). These studies should be considered highly uncertain, however, because they necessarily make large assumptions about the amount of zooplankton mortality occurring in the deep ocean, where it has never been directly quantified.

Results from our study, which does include mortality at depth, suggests that active transport is a quantitatively important, but never dominant component of the BCP, in line with results from a recent global estimate derived from a combination of satellite remote-sensing products and modeling approaches (Archibald et al., 2019).

One aspect of the BCP that our current euphotic-zone only simulations do not address is sequestration efficiency in the mesopelagic (Kwon et al., 2009; Marsay et al., 2015; Buesseler and Boyd, 2009). It is reasonable

to surmise that the remineralization length scale will vary for different BCP pathways and be regionally variable as well. With gravitational flux, typically ~50% of particles will sink 100 m beneath the euphotic zone before remineralization, although remineralization length scales are highly variable and the spatial patterns are poorly understood (Buesseler and Boyd, 2009; Marsay et al., 2015). Meanwhile, vertically-migrating zooplankton typically reside at depths of 200 – 600 m during the day and hence respire the majority of their carbon dioxide at this depth

(Bianchi et al., 2013b), although it is unclear how the inclusion of mortality at depth into our understanding of active transport will affect the overall depth of penetration of actively transported carbon into the deep ocean. Stukel et al. (2018b), suggested that subducted particles in the southern CCE are mostly remineralized near the base of the euphotic zone with little penetration into the mesopelagic, although in regions with deep convective mixing, signatures of subduction show substantial transport into the deep ocean (Omand et al., 2015; Llort et al., 2018).

Boyd et al. (2019) surmised that active transport may have the greatest sequestration efficiency, followed by vertical mixing, then gravitational flux, although their synthesis was only able to draw from the few studies that have quantified these processes and they note that determining the sensitivities of sequestration efficiencies to environmental drivers is crucial to predicting climate change impacts on marine carbon sequestration. We believe that future incorporation of our model ensemble approach into three-dimensional coupled modeling frameworks

could be an important step forward in understanding the magnitude, and uncertainty in these processes.

### 4.2. Data-assimilating biogeochemical models

Implicit to our OEP$_{MCMC}$ approach is the philosophical realization that our model (like all biogeochemical models) oversimplifies an incredibly complex system. Hence, we accept that no single solution set will accurately simulate all aspects of the BCP. Instead, we proposed a mechanistic-probabilistic approach that explicitly

investigates the ecosystem uncertainty. This contrasts with some other data-assimilation approaches (e.g., gradient-based methods including the variational adjoint, Schartau et al., 2001; Friedrichs et al., 2007; Lawson et al., 1995) that seek to find a single solution that minimizes model-data misfit. While the variational-adjoint approach is computationally efficient and allows objective determination of a single solution that can then be used for high-resolution simulations (Mattern et al., 2017), our work shows that very different parameter sets can result in similar

cost function values, despite generating distinctly different model outputs.

Our approach has similarities with other biogeochemical model ensemble approaches. For instance, Doron et al. (2013) used an ensemble Kalman filter algorithm to assimilate surface chlorophyll data and determine regional variability in biogeochemical parameters for a simple ecosystem model. Gharamti et al. (2017a; 2017b) used a





modified approach to simultaneously estimate model parameters and state variable distributions to enable reasonably accurate ensemble predictions of modeled processes.   These Kalman filter approaches are widely used in physical sciences for state estimation, re-analyses, and prediction purposes, although the data-assimilating state variable updates sacrifice true dynamical consistency.   Meier et al. (2011) used dynamically consistent model ensembles generated from three different biogeochemical models forced with four climate projections and three different nutrient loading scenarios to investigate increasing hypoxia in the Baltic Sea.   Garnier et al. (2016) used a

probabilistic version of the NEMO/PISCES model to generate a 60-member ensemble simulation of chlorophyll in the North Atlantic that accounts for uncertainties in biogeochemical parameters and sub-grid scale processes.   Gal et al. (2014) conducted a single model ensemble approach similar to ours in which they perturbed the most sensitive parameters in their model to investigate whether trends associated with different nutrient loading scenarios were consistent across the ensemble, although their approach did not use data assimilation to determine the different

parameter values used.   Ramondenc et al. (2020) used the statistical model check engine to assimilate laboratory and *in situ* observations to probabilistically constrain parameters associated with scyphozoan growth and degrowth. Anugerahanti et al. (2018) conducted a model ensemble approach in which, rather than modifying parameter values, they modified the functional form of key transfer functions associated with nutrient uptake, grazing, and mortality while simulated chlorophyll, nutrients, and primary production at 5 time-series sites.   They discovered that the

model was especially sensitive to modifications to the grazing and mortality functions.   A further study (Anugerahanti et al., 2020) simultaneously perturbed physical circulation fields and the biogeochemical model and found that results were most sensitive to variability in the biological model.   The result of these ensemble approaches is a probabilistic estimate of model outputs that (hopefully) accurately reflects true uncertainty in the system.   Our OEP$_{MCMC}$ approach, by utilizing field data to automate the choice of parameter sets to be used in the

model ensemble, allows us to generate one million different dynamically consistent model realizations that each fit the available data, while simultaneously exploring different regions of the solution space with regard to uncertainties in all of the modeled parameters.   We consider this to be a reasonable tradeoff for the increased computational expense of our approach (relative to the variational adjoint or Kalman filter approaches), while noting that each approach has distinct advantages or disadvantages for different applications.

An additional novelty of our study is the variety of different data types assimilated into the model (30 different rate and standing stock measurement types).   Most data-assimilating biogeochemical models only incorporate data associated with nutrients and/or surface chlorophyll and other remotely-sensed parameters (e.g., Xiao and Friedrichs, 2014b; Mattern et al., 2014; Wang et al., 2012).   The incorporation of multiple data types spanning trophic levels and biogeochemical processes is important to model validation, because models can often reasonably

simulate time series of one particular variable, with unrealistic dynamics of associated trophic levels.   Ciavatta et al. (2014) found that assimilation of light attenuation coefficient data improved model prediction of light attenuation coefficient data, but did not improve model estimates of surface chlorophyll, and even degraded model performance in some regions.   Furthermore, assimilation of only noisy standing stock data can lead to model overfitting and inability to retrieve accurate model parameters, even in an idealized model (Löptien and Dietze, 2015).   The few

studies that have attempted to incorporate many measurement types have focused on nutrient and phytoplankton parameters.   For instance, Kim et al. (2021) assimilated standing stock data associated with 9 model compartments along with net primary production and bacterial production into a model of an Antarctic coastal ecosystem but incorporated no metazoan zooplankton data.   In a model simulating three distinct open ocean regions, Luo et al.





(2010) incorporated only one zooplankton parameter (mesozooplankton biomass) amongst 17 assimilated data types, mostly associated with non-living compartments. By contrast, we incorporate an extensive suite of group-specific protistan grazing rate measurements and biomass and grazing rate measurements for each of our 4 metazoan zooplankton groups. While these provide realistic bounds within which zooplankton dynamics can vary, zooplankton parameters still remain among the least constrained parameters in our model due to the difficulty of

making zooplankton rate measurements (e.g., the paucity of grazing measurement relative to net primary production) and the fact that most zooplankton measurements (derived from net tows) inherently integrate over broad depth ranges. The weak constraints on zooplankton processes are particularly important in light of multiple studies that have shown that even subtle changes in grazing formulations can fundamentally alter biogeochemical behaviors of models (Sailley et al., 2015; Gentleman and Neuheimer, 2008; Schartau et al., 2017; Chenillat et al., 2021; Sailley et al., 2013; Prowe et al., 2012) and the crucial roles of metazoan zooplankton for multiple pathways

of the BCP (Buitenhuis et al., 2006; Steinberg and Landry, 2017).

### 4.3. Future directions

       We have highlighted some of the insight about the BCP that can be gleaned from our ensemble data assimilation approach. However, as noted previously, there are many limitations associated with using a simplified one-dimensional physical framework, and indeed a large portion of our study goal was to set the stage for more

advanced uses of NEMURO$_{BCP}$ and OEP$_{MCMC}$. One obvious future step is to incorporate NEMURO$_{BCP}$ into three-dimensional circulation models. Although NEMURO$_{BCP}$ was originally written in Matlab, we are currently adapting it to Fortran compatible with circulation models such as ROMS, HYCOM, and MITgcm. Three-dimensional NEMURO$_{BCP}$ simulations may take different forms. One approach would be to use different parameter sets from the data ensemble in independent model runs, to conduct three-dimensional global biogeochemical model

ensembles. Notably, our different parameter sets are equally supported by assimilated field data, yet some predict very different ecosystem states (e. g., they vary in relative proportion of large/small phytoplankton, in turnover times for biota, in partitioning of organic matter between the particulate and dissolved phase, etc.). This ensemble modeling approach would thus allow quantification of BCP uncertainties in four dimensions. An alternate approach would be to use parameter distributions from one-dimensional simulations as prior estimates of parameters for data-

assimilation in a three-dimensional model. These prior estimates of each parameter (and the parameter covariance matrix) could be incorporated into the cost function for many different data-assimilation approaches. Comparison to satellite-observed or *in situ* time-series data would add powerful additional constraints on parameter values.

       Another future use of the ensemble approach would be to simulate the results of specific Lagrangian experiments. In the current study, we developed an ensemble of plausible parameter sets that could be used for

global ensemble models in the future or as prior distributions for future studies, while also assessing the uncertainty in parameter values. These goals informed our decision to conduct a joint parameter estimation that simultaneously utilized data from all available experiments (rather than estimating different parameter values for each experiment or each region). To simulate ecosystem dynamics for a specific experiment as accurately as possible, one would need to treat initial conditions and boundary values as unknown values to be determined during the optimization

procedure. As such, the cost function should formally be defined as a function of these unknown values: $J(I_C, B_V, F, P)$ where $I_C$ represents the initial conditions (all state variables, all depths), $B_V$ is the boundary values (i.e., values of the state variables at the bottom boundary of the model), $F$ is the physical forcing, and $P$ is the parameter set. While





this introduces a large number of additional unknown variables to solve for, it also justifies use a more stringent cost function (e.g., the likelihood function). Thus to use NEMURO$_{BCP}$ to model a specific Lagrangian experiment (e.g.,

time-varying conditions during the North Pacific EXPORTS Lagrangian study, Siegel et al., 2021), we recommend treating our results for estimated global ranges of parameters as prior values in a Bayesian analysis to simultaneously constrain $I_C$, $B_V$, $F$, and $P$ for that Lagrangian experiment.

In the current study, we incorporated a broad suite of standing stock and rate measurements spanning nutrients, phytoplankton, zooplankton, and non-living organic matter, because our goal was to simultaneously constrain all

parameters in the model while investigating overall uncertainty in model outputs. However, Loptien & Dietze (2015) noted that specific parameters and processes can be better constrained if only the most relevant type of data is included. We thus suggest that targeted choice of data types to assimilate could allow the use of OEP$_{MCMC}$ for investigation of specific processes that are difficult to directly measure *in situ*. For instance, zooplankton mortality at depth has been hypothesized to be a potentially important component of the BCP (Kelly et al., 2019; Hernández-

León et al., 2019), but estimates of zooplankton mortality at depth are typically derived from either allometric relationships between zooplankton size and life span or estimates of mortality made in the upper ocean (Brett and Groves, 1979; Hirst and Kiørboe, 2002; Ohman and Hirche, 2001). By incorporating only the data sources that offer the most constraint on zooplankton parameters (e.g., biomass and grazing rates of each zooplankton group), it may be possible to better constrain the fraction of mortality occurring in the deep ocean.

NEMURO$_{BCP}$ was built off of the NEMURO family of models (Kishi et al., 2007), and here we only added extra state variables essential for modeling BCP pathways from the euphotic zone into the mesopelagic. There are, of course, multiple additional processes that are important to simulating marine biogeochemistry and the BCP that are currently absent. Some additional processes that we consider priorities and plan to implement in future versions of NEMURO$_{BCP}$ include variable stoichiometry of organic matter, N$_2$ fixation, and additional realism in the

microbial community. Elemental stoichiometry (e.g., C:N:P) can vary substantially between different organic pools and across the different BCP pathways (Hannides et al., 2009; Singh et al., 2015), is predicted to change as a result of ocean acidification and/or increased temperature and stratification (Oschlies et al., 2008; Riebesell et al., 2007), and can affect the balance between carbon sequestration and nutrient supply and regeneration leading to potentially enhanced carbon sequestration and growing oxygen minimum zones in a future ocean (Michaels et al., 2001;

Oschlies et al., 2008; Riebesell et al., 2007). Adding variable stoichiometry to NEMURO$_{BCP}$ is simple but will require the addition of state variables associated with each model compartment that is allowed to vary in its elemental ratios, with substantial added computational costs. N$_2$ fixation is simultaneously a source of new production in the absence of upwelling and a process that can substantially alter elemental stoichiometry in the open ocean. It could be introduced to the model through a state variable(s) simulating diazotrophs (Hood et al., 2001) or

through implicit parameterization (Ilyina et al., 2013). NEMURO$_{BCP}$ might also benefit from added realism in microbial dynamics. The roles of heterotrophic bacteria in particle remineralization are currently included implicitly in the model. Explicit simulation of bacterial biomass and processes such as colonization of particles, microbial hotspots on sinking particles, production of hydrolytic enzymes, quorum sensing, and predator-prey dynamics with protists have the potential to more accurately simulate feedbacks that affect remineralization length scales in the

ocean (Robinson et al., 2010; Simon et al., 2002; Mislan et al., 2014). Additionally, the model currently includes only two phytoplankton, which were explicitly identified as diatoms and non-diatoms in this data-assimilation exercise. The latter category subsumes a wide variety of different phytoplankton taxa into a group with transfer





functions designed to simulate picophytoplankton (especially cyanobacteria). It thus excludes the presence of mixotrophs, which are abundant and diverse bacterivores in the open ocean, can survive low-nutrient and low-light conditions by supplementing their nutritional budget with phagotrophy, and may have distinctly different biogeochemical roles due to their decreased reliance on dissolved nutrients (Stoecker et al., 2017; Jones, 2000).

## 5. Conclusions

The data assimilation approach utilized here is a computationally feasible method for incorporating a diverse suite of *in situ* measurements to objectively define parameter sets for ensemble modeling of the BCP. The 30 data types assimilated in this study improve constraints on ecosystem dynamics. However, some parameters, especially those related to metazoan zooplankton, remain poorly constrained by available data, despite assimilation of 8 data types explicitly representing metazoan zooplankton rates and standing stocks. This likely results from a combination of the inherently patchy nature of many mesozooplankton populations (i.e., high measurement uncertainty) and the vertically integrated nature of zooplankton net tows which obscures simple relationships between predator abundance, prey abundance, and grazing rates.

The three BCP pathways were spatiotemporally variable across four study regions. Despite a very simple physical framework, distinct patterns were identified. Active transport was only a dominant contributor to the BCP in the CRD, where simulations predicted it to be responsible for 20-40% of export from the euphotic zone. Near the subtropical front of the Southern Ocean and in upwelling-influenced regions of the CCE, both gravitational flux and vertical mixing were important components of the BCP, with the relative importance of the two determined more by differences between parameter sets, than by differences between the conditions experienced during specific Lagrangian experiments. In offshore oligotrophic regions of the CCE and the GoM >80% of export was usually attributable to gravitational flux, although mixing dominated in a few experiments.

Our ensemble approach highlights uncertainties around model estimates of the BCP that arise from imprecisely defined parameters. Indeed, variability in many aspects of the BCP is as large comparing different (realistic) parameter sets within a specific location as it is across regions as distinctly different as the oligotrophic GoM and coastal CCE. Notably, different realistic parameter sets from our ensembles predict very different export efficiencies (and hence magnitudes of the gravitational pump) despite similar net primary production. This suggests that model validation against net primary production (or sea surface chlorophyll) data is insufficient to validate model skill in simulating BCP variability. The explicit representation of thorium and nitrogen isotope dynamics in NEMURO$_{BCP}$ should aid in future model validation efforts.



**Code Availability**

The core NEMURO$_{BCP}$ code is available on GitHub at: https://github.com/mstukel/NEMURO_BCP. The code

necessary to run the objective ensemble parameterization procedure can be found at:

https://github.com/mstukel/OEP_MCMC_NEMURObcp.

**Data Availability**

Field data used in this manuscript is available on either the CCE LTER Datazoo repository

(https://oceaninformatics.ucsd.edu/datazoo/catalogs/ccelter/datasets) or the Biological and Chemical Oceanography
Data Management Office repository: https://www.bco-dmo.org/project/834957, https://www.bco-
dmo.org/project/819488, and https://www.bco-dmo.org/project/754878. For ease of access it is also included in
Supp. Tables S2-S4. The data file containing all model outputs (from all ensembles) is too large to deposit but can
be generated from the code on GitHub. A summarized version (every 1000$^{th}$ iteration) is included as Supp. Table

S5, summary statistics are given in Supp. Table S1, with the correlation and covariance matrices given in Supp.
Table S6.

**Author Contribution**

MRS developed NEMURO$_{BCP}$ model and performed the simulations. MD led SalpPOOP project. MRL led
BLOOFINZ and FluZIE projects. MRS prepared the manuscript with contributions from all co-authors.

**Acknowledgements**

We thank the captains and crews of the R.V.s Melville, Revelle, Knorr, Thompson, Sikuliaq, and Tangaroa and the
NOAA ship Nancy Foster. We are also grateful to our many collaborators in the CCE LTER, BLOOFINZ-GoM,
CRD FluZIE, and SalpPOOP Projects, especially Mark Ohman, Karen Selph, Thomas Kelly, Ralf Goericke, Scott
Nodder, Andres Gutierrez-Rodriguez, Jeff Krause, and Karl Safi. This proposal was funded by U. S. National

Science Foundation awards OCE1756610 and 1851347 to M.R.S.; OCE-0826626 to M.R.L; and OCE-0417616,
OCE-1026607, OCE-1637632, and OCE-1614359 to the CCE LTER Program. It was also funded by the National
Oceanic and Atmospheric Administration's RESTORE Program Grant (Project Title: Effects of nitrogen sources
and plankton food-web dynamics on habitat quality for the larvae of ABT in the GoM; under federal funding
opportunity NOAA-NOSNCCOS-2017-2004875) and by the Ministry for Business, Innovation and Employment

(MBIE) of New Zealand, by NIWA core programs Coast and Oceans Food Webs (COES), Ocean Flows (COOF),
and by the Royal Society of New Zealand Marsden Fast-track award to M.D.

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

**Table 1. State variables in NEMURO_BCP**

| Abbreviation | Description | Units |
|---|---|---|
| **Core model** | | |
| SP | Small (non-diatom) phytoplankton | mmol N m$^{-3}$ |
| LP | Large phytoplankton (diatoms) | mmol N m$^{-3}$ |
| SZ | Small (protistan) zooplankton | mmol N m$^{-3}$ |
| LZ$_{RES}$ | <1-mm epipelagic-resident mesozoopankton | mmol N m$^{-3}$ |
| LZ$_{DVM}$ | <1-mm diel-vertically-migrating mesozooplankton | mmol N m$^{-3}$ |
| PZ$_{RES}$ | >1-mm epipelagic-resident mesozoopankton | mmol N m$^{-3}$ |
| PZ$_{DVM}$ | >1-mm diel-vertically-migrating mesozooplankton | mmol N m$^{-3}$ |
| NO3 | Nitrate | mmol N m$^{-3}$ |
| NH4 | Ammonium | mmol N m$^{-3}$ |
| PON | Slowly-sinking detritus | mmol N m$^{-3}$ |
| LPON | Rapidly-sinking detritus | mmol N m$^{-3}$ |
| DON | Labile dissolved organic nitrogen | mmol N m$^{-3}$ |
| DONref | Refractory dissolved organic nitrogen | mmol N m$^{-3}$ |
| SI | Silicic acid | mmol Si m$^{-3}$ |
| OP | Slowly-sinking opal (biogenic silica) | mmol Si m$^{-3}$ |
| LOP | Rapidly-sinking opal (biogenic silica) | mmol Si m$^{-3}$ |
| CHL$_{PS}$ | Chlorophyll associated with small phytoplankton | mg Chl *a* m$^{-3}$ |
| CHL$_{PL}$ | Chlorophyll associated with large phytoplankton | mg Chl *a* m$^{-3}$ |
| OXY | Dissolved oxygen | mmol O m$^{-3}$ |
| **Carbon submodule** | | |
| DIC | Dissolved inorganic carbon | mmol C m$^{-3}$ |
| ALK | Alkalinity | mmol m$^{-3}$ |
| **$^{234}$Thorium submodule** | | |
| dTh | Dissolved $^{234}$Th | dpm L$^{-1}$ |
| SP$_{Th}$ | $^{234}$Th adsorbed to small phytoplankton | dpm L$^{-1}$ |
| LP$_{Th}$ | $^{234}$Th adsorbed to large phytoplankton | dpm L$^{-1}$ |
| SZ$_{Th}$ | $^{234}$Th adsorbed to small zooplankton | dpm L$^{-1}$ |
| LZRES$_{Th}$ | $^{234}$Th adsorbed to LZRES | dpm L$^{-1}$ |
| LZDVM$_{Th}$ | $^{234}$Th adsorbed to LZDVM | dpm L$^{-1}$ |
| PZRES$_{Th}$ | $^{234}$Th adsorbed to PZRES | dpm L$^{-1}$ |
| PZDVM$_{Th}$ | $^{234}$Th adsorbed to PZDVM | dpm L$^{-1}$ |
| PON$_{Th}$ | $^{234}$Th adsorbed to slowly-sinking detritus | dpm L$^{-1}$ |





| | | |
|---|---|---|
| LPON$_{Th}$ | $^{234}$Th adsorbed to rapidly-sinking detritus | dpm L$^{-1}$ |
| **Nitrogen isotope submodule** | | |
| SP$_{N15}$ | $^{15}$N in small phytoplankton | mmol $^{15}$N m$^{-3}$ |
| LP$_{N15}$ | $^{15}$N in large phytoplankton | mmol $^{15}$N m$^{-3}$ |
| SZ$_{N15}$ | $^{15}$N in small zooplankton | mmol $^{15}$N m$^{-3}$ |
| LZRES$_{N15}$ | $^{15}$N in LZRES | mmol $^{15}$N m$^{-3}$ |
| LZDVM$_{N15}$ | $^{15}$N in LZDVM | mmol $^{15}$N m$^{-3}$ |
| PZRES$_{N15}$ | $^{15}$N in PZRES | mmol $^{15}$N m$^{-3}$ |
| PZDVM$_{N15}$ | $^{15}$N in PZDVM | mmol $^{15}$N m$^{-3}$ |
| NO$_{N15}$ | $^{15}$N in nitrate | mmol $^{15}$N m$^{-3}$ |
| NH$_{N15}$ | $^{15}$N in ammonium | mmol $^{15}$N m$^{-3}$ |
| PON$_{N15}$ | $^{15}$N in slowly-sinking detritus | mmol $^{15}$N m$^{-3}$ |
| LPON$_{N15}$ | $^{15}$N in rapidly-sinking detritus | mmol $^{15}$N m$^{-3}$ |
| DON$_{N15}$ | $^{15}$N in labile DON | mmol $^{15}$N m$^{-3}$ |
| DONREF$_{N15}$ | $^{15}$N in refractory DON | mmol $^{15}$N m$^{-3}$ |





**Figures**

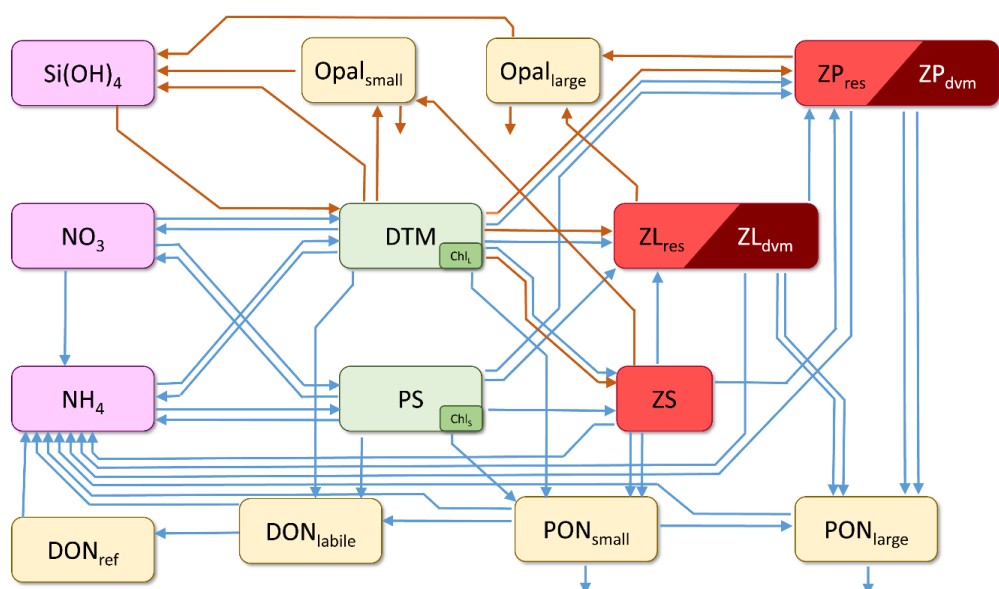

**Figure 1 -** Schematic depiction of core NEMURO$_{BCP}$ model. Arrows show transfer functions (orange = Si flux; blue = N flux). Rectangles show state variables (SiOH$_3$ = silicic acid; NO$_3$ = nitrate; NH$_4$ = ammonium; Opal$_{small}$ =
small biogenic silica; Opal$_{large}$ = large biogenic silica; DON$_{ref}$ = refractory dissolved organic nitrogen; DON$_{labile}$ = labile dissolved organic nitrogen; PON$_{small}$ = small detritus; PON$_{large}$ = large detritus; DTM = diatoms; PS = small phytoplankton; Chl$_l$ = diatom chlorophyll; chl$_s$ = small phytoplankton chlorophyll; ZS = protistan zooplankton; ZL$_{res}$ = <1-mm metazoan zooplankton that are resident in the euphotic zone; ZL$_{dvm}$ = <1-mm diel-vertically-migrating metazoan zooplankton; ZP$_{res}$ = >1-mm metazoan zooplankton that are resident in the euphotic zone; ZP$_{dvm}$ = >1-mm
diel-vertically-migrating metazoan zooplankton. Oxygen is also a state variable but is not shown in this figure.





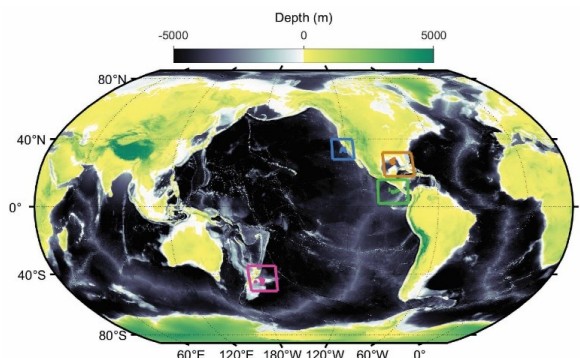

**Figure 2** – Locations of our in situ Lagrangian experiments (blue = California Current Ecosystem, Brown = Gulf of Mexico, Green = Costa Rica Dome, Magenta = Chatham Rise).





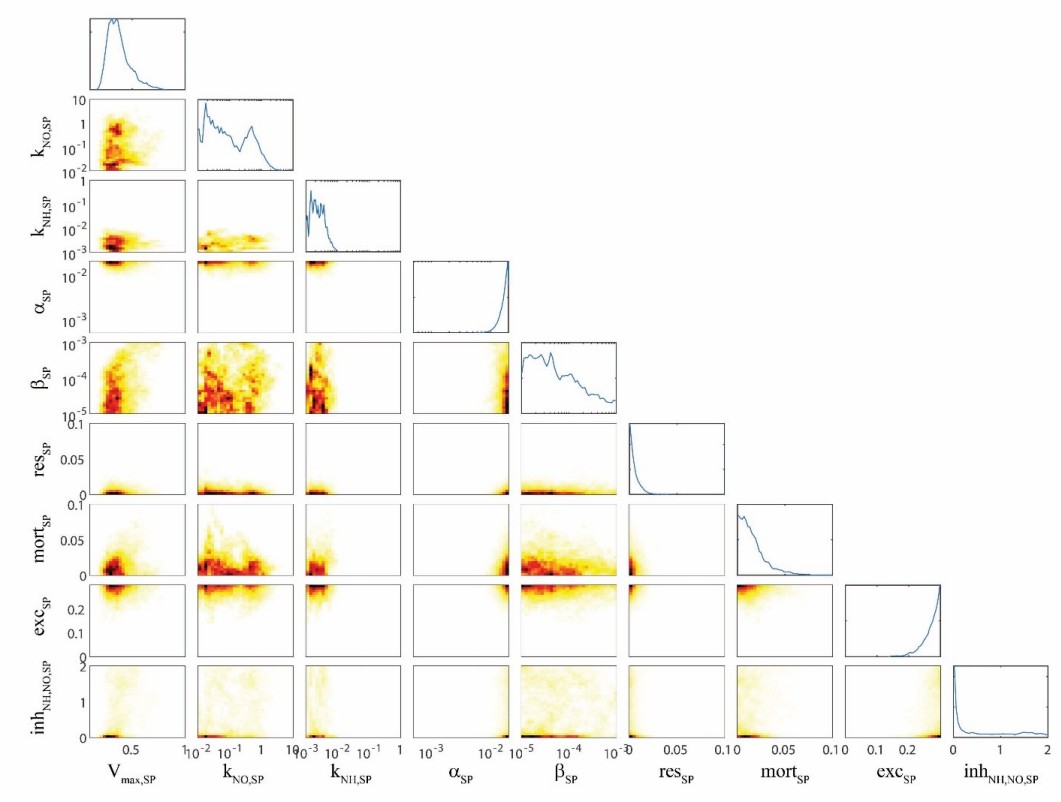


**Figure 3** – OEP$_{MCMC}$ parameter distributions for bottom-up control of small phytoplankton. Line plots on top are probability density functions for individual parameters (see bottom for label and axes). Colored plots are heat maps showing joint parameter distributions. Parameters are: maximum growth rate at 0°C (V$_{max,SP}$, units = d$^{-1}$), half-saturation constant for nitrate uptake (K$_{NO,SP}$, mmol N m$^{-3}$), half-saturation constant for ammonium uptake (K$_{NH,SP}$, mmol N m$^{-3}$), initial-slope of the photosynthesis-irradiance curve (α$_{SP}$, m$^2$ W$^{-1}$ d$^{-1}$), photoinhibition parameter (β$_{SP}$, m$^2$ W$^{-1}$ d$^{-1}$), respiration rate at 0°C (res$_{SP}$, d$^{-1}$), linear mortality term at 0°C (mort$_{SP}$, d$^{-1}$), excretion parameter (exc$_{SP}$, unitless), ammonium inhibition of nitrate uptake (inh$_{NH,NO,SP}$, m$^3$ mmol N$^{-1}$).
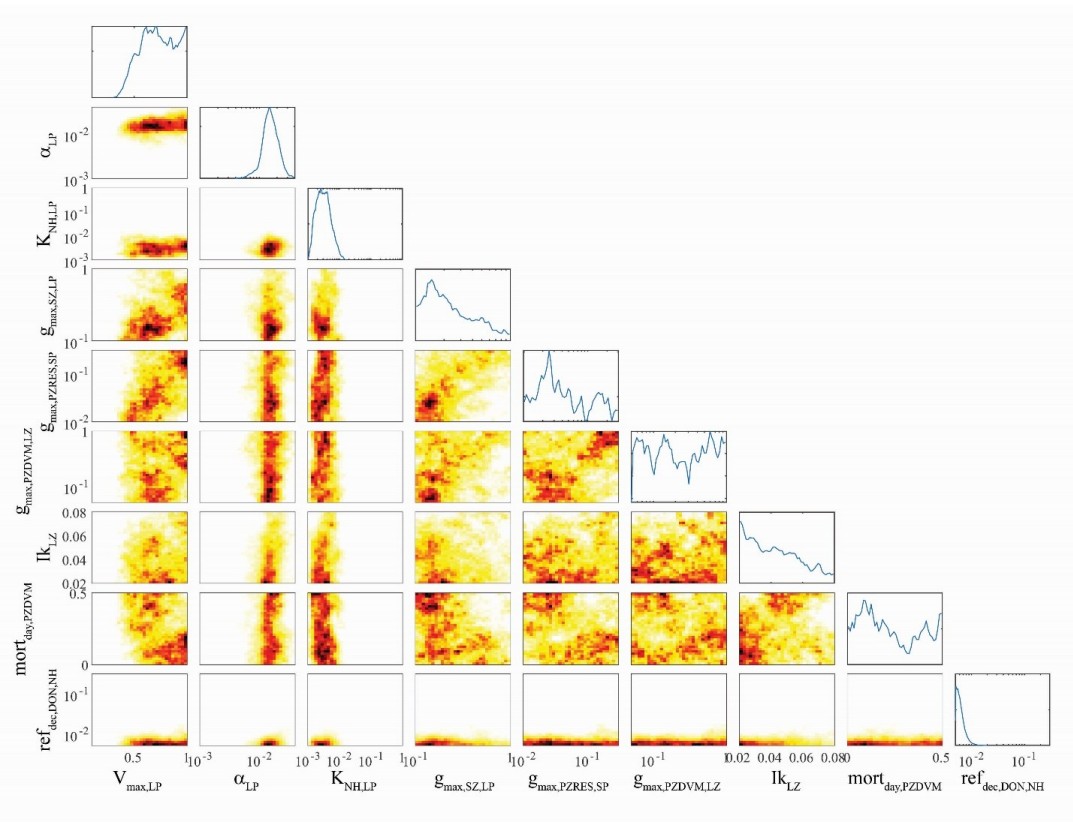

**Figure 4** – OEP$_{MCMC}$ parameter distributions for large phytoplankton and some other other model processes. Line plots on top are probability density functions for individual parameters (see bottom for label and axes). Colored plots are heat maps showing joint parameter distributions. Parameters are: maximum growth rate at 0°C (V$_{max,LP}$, units = d$^{-1}$), initial-slope of the photosynthesis-irradiance curve ($\alpha_{LP}$, m$^2$ W$^{-1}$ d$^{-1}$), half-saturation constant for NH$_4^+$ uptake (K$_{NH,LP}$, mmol N m$^{-3}$), maximum grazing rate of small zooplankton on large phytoplankton (g$_{max,SZ,LP}$, d$^{-1}$).

maximum grazing rate of large (>1-mm) epipelagic-resident mesozooplankton on small phytoplankton (g$_{max,PZRES,SP}$, d$^{-1}$), maximum grazing rate of large (>1-mm) vertically-migrating mesozooplankton on small (<1-mm) mesozooplankton (g$_{max,PZDVM,LZ}$, d$^{-1}$), the Ikeda respiration parameter for small (<1-mm) mesozooplankton, daytime mortality rate for small (<1-mm) vertically-migrating mesozooplankton (mort$_{day,LZDVM}$, m$^3$ mmol N$^{-1}$ d$^{-1}$), remineralization rate of DON to NH$_4^+$ (ref$_{dec,DON,NH}$, d$^{-1}$).






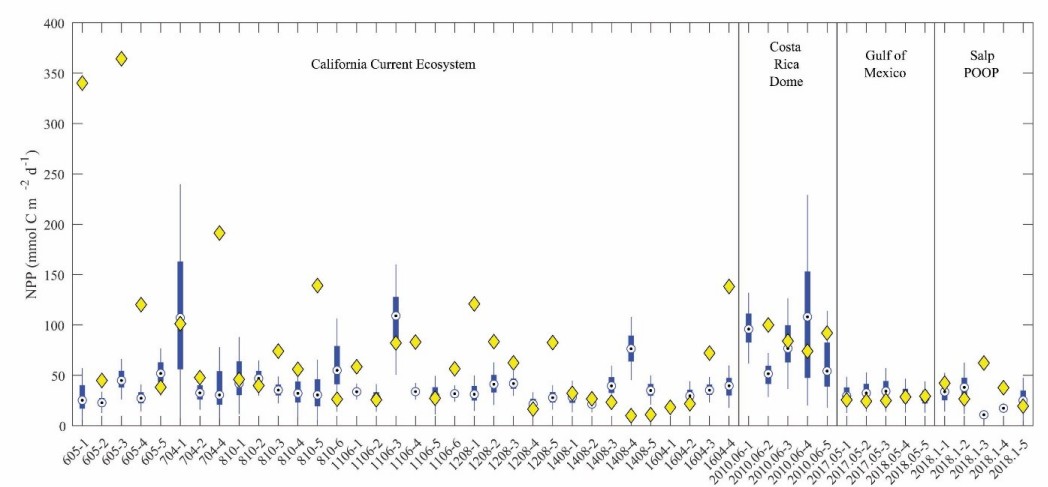

**Figure 5** – Model-data net primary production comparison. Blue box plots show model results for each simulated Lagrangian experiment, with whiskers extending to 95% confidence limits. Yellow diamonds show observations from Lagrangian experiments.

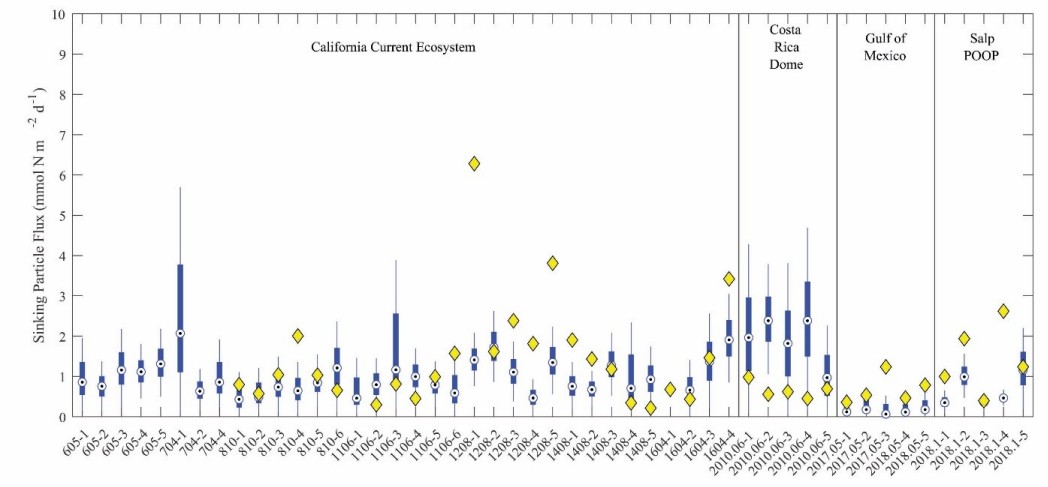

**Figure 6** – Model-data sinking particle export comparison. Blue box plots show model results for each simulated Lagrangian experiment, with whiskers extending to 95% confidence limits. Yellow diamonds show observations from sediment trap deployments (no observations were available for 9 experiments).



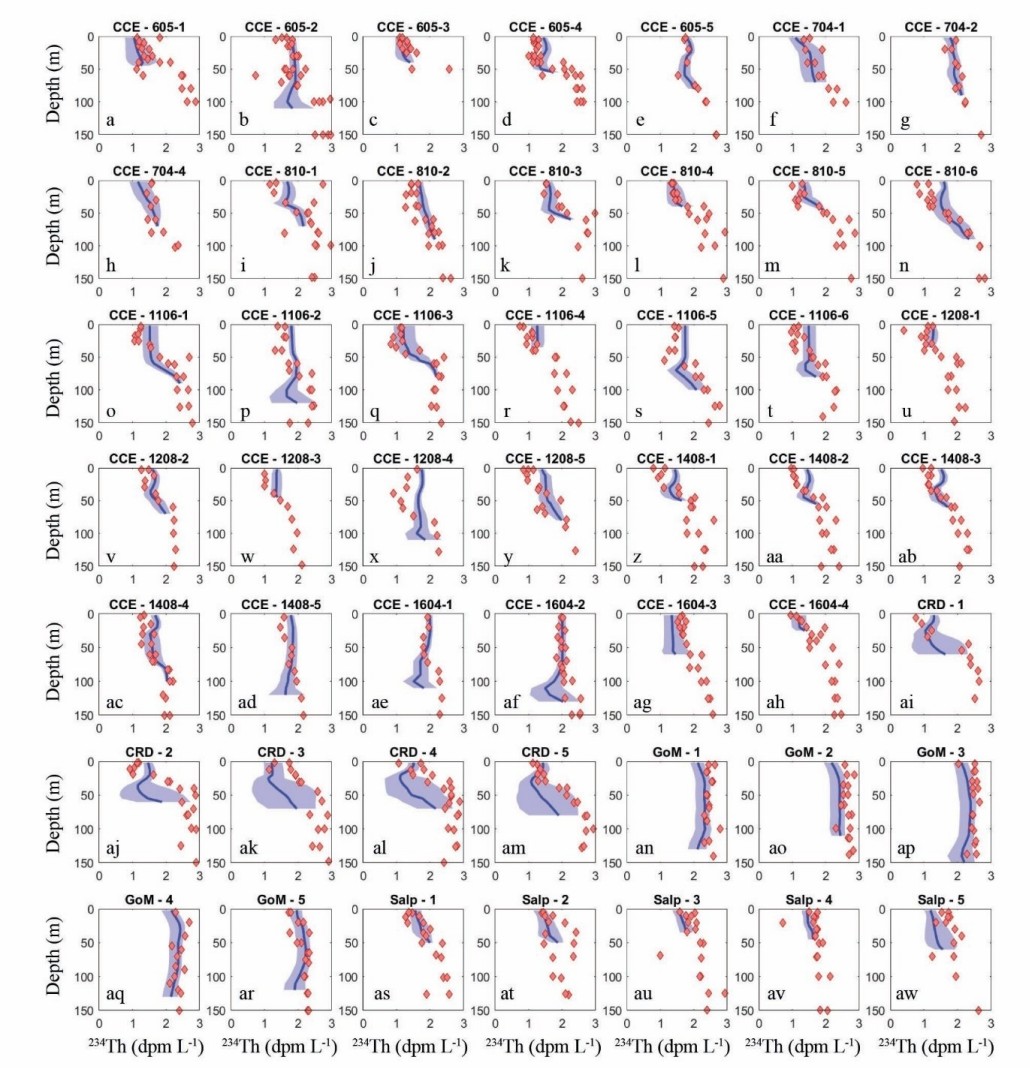

**Figure 7** – Model-data water-column ²³⁴Th activity comparison. Dark blue lines show mean vertical profile of ²³⁴Th
activity from MCMC model simulations with lighter blue shading indicating 95% C.I. Red diamonds show
observations. Each panel is for a separate Lagrangian experiment.



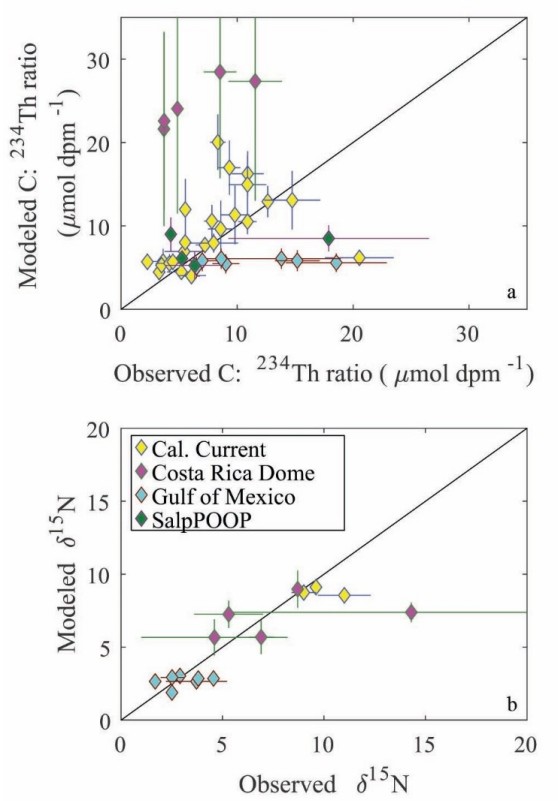

**Figure 8** – Model-data comparison of C:$^{234}$Th ratio (a) and δ$^{15}$N of sinking particles. Color indicates region. Error bars are ±1 standard deviation. Black line is the 1:1 line. Observations are derived from sediment trap measurements.

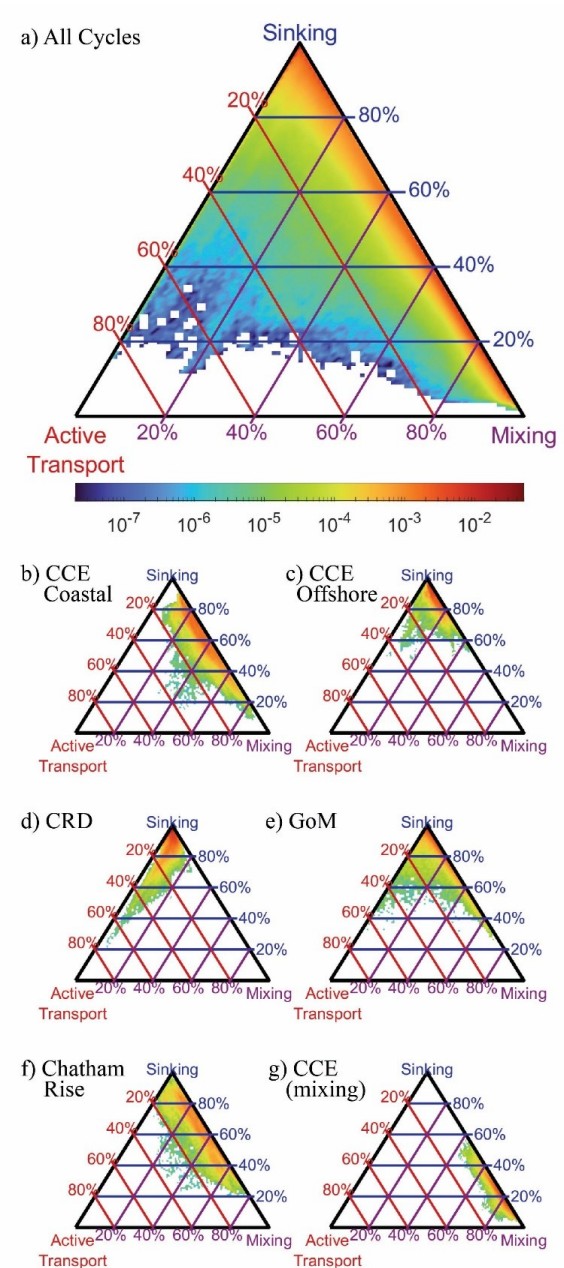

**Figure 9** – Triangle diagrams showing the proportion of export due to each biological carbon pump pathway.
Locations near the upper apex of the triangle indicated dominance by sinking particles, locations near the bottom left indicate dominance by active transport, locations near the bottom right show dominance by mixing. Colors represent the proportion of total model simulations with export patterns falling within a specific proportion of different export pathways. Lines indicated contours showing a constant proportion of one BCP pathway (i.e., red lines are constant proportions of active transport, blue lines are constant proportions of gravitational flux, and purple lines are constant proportions of mixing flux). a) results for all simulations, b) results for a typical CCE coastal site (1604-3), c) typical CCE oligotrophic site (1408-5), d) typical Costa Rica Dome site (CRD-1), e) typical Gulf of Mexico site (GoM-5), f) typical Chatham Rise site (Salp-5), g) example of a CCE site (0605-3) dominated by mixing flux.

**Figure 10** – Heatmaps of active transport (a), active transport due to excretion in the deep ocean (b), active transport due to mesozooplankton
mortality at depth (c), the fraction of active transport that was due to mortality at depth (d), and the daytime specific mortality experienced by
mesozooplankton at their mesopelagic resting depths (e), all as a function of the total biomass of vertically-migrating mesozooplankton (i.e., sum
of both size classes). Black lines and equations in a, b, c, and d were determined from ordinary least squares regressions of log-transformed data
(see text for regression statistics). (f) shows the probability density function for the fraction of large (>1 mm) mesozoolpankton mortality
experienced during the day at their mesopelagic resting depths. (g) and (h) show normalized histograms of $\log_{10}$-transformed zooplankton
biomass and active transport, respectively. Dashed blue line is small epipelagic-resident zooplankton, solid blue is small DVM zooplankton,
dashed red is large epipelagic-resident zooplankton, solid red is large DVM zooplankton. (i) shows the fraction of active transport mediated by
large mesozooplankton (>1 mm) as a function of their fraction of total vertically-migrating mesozooplankton biomass. Dashed gray line is the
1:1 line.





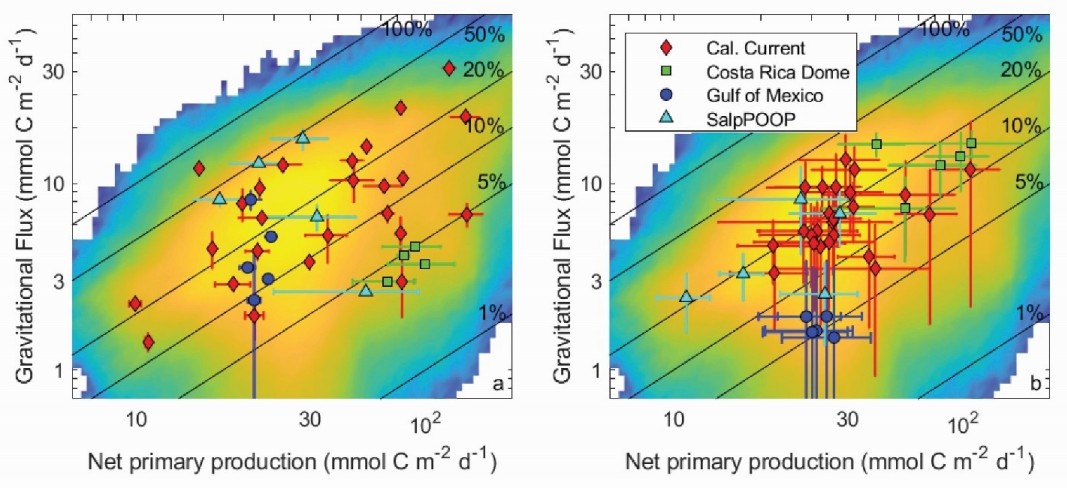


**Figure 11** – Gravitational flux as a function of net primary production for in situ data (a) and model results (b). Averages and standard deviations are shown for individual Lagrangian experiments. Nitrogen-based model results were converted to carbon units assuming a C:N ration of 106:16 (mol:mol). Background in both figures is a heatmap of all model results (i.e., all Lagrangian experiments and all parameter sets). Solid black lines show

contours of constant *e*-ratio (=gravitational flux / net primary production).

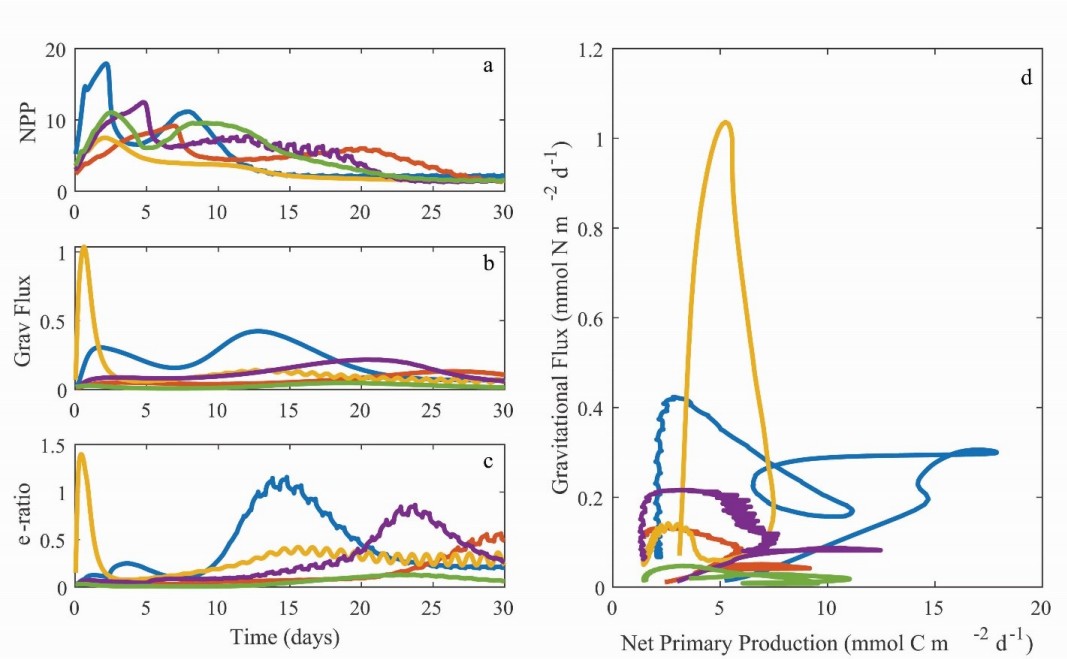

**Figure 12** – Temporal variability in net primary production (a, mmol C m$^{-2}$ d$^{-1}$), gravitational flux (b, mmol N m$^{-2}$ d$^{-1}$), and export efficiency (c, unitless with a C:N conversion ratio of 106:16 mol:mol), along with a phase-space plot

depicting the same data (d). All plots are from Lagrangian experiment 1604-3 (CCE upwelling region). Different colors are for simulations with ensemble parameter sets 2×10$^5$, 4×10$^5$, 6×10$^5$, 8×10$^5$, or 10$^6$.