# Peer review of "Quantifying biological carbon pump pathways with a data-constrained mechanistic model ensemble approach"

_Biogeosciences, 2022_

## Author Response (AR1)

**Associate Editor decision: Publish subject to minor revisions (review by editor)**
**Comments to the author**:
Dear Michael Stukel and co-authors,
I have read the answers you provided to the comments of the two reviewers. Based on that, I invite you to submit a revised version of your work that incorporates all the changes mentioned in your answer. Please provide an annotated revised manuscript clearly highlighting all the changes made.
Thanks a lot for your efforts in revising the manuscript,
Kind regards,
Marilaure Gregoire

**We greatly appreciate this opportunity to revise and resubmit our manuscript. Please see our point-by-point responses to the reviewers below.**

**Reviewer #1: Vassilios Vervatis**

Title: Quantifying biological carbon pump pathways with a data-constrained mechanistic model ensemble approach.

by Stukel et al.,

General comments:

The study investigates the pathways of biological carbon pump, performing ensemble simulations of biogeochemical model parameterizations, constrained by data assimilation with the use of several data types obtained from Lagrangian experiments.

The ms. is well written and well structured, being very informative for the processes controlling BCP pathways. The idea of using an ensemble-based approach to quantify model parameter uncertainties and constrain them by data assimilation is innovative and the general approach is meaningful.

I am not an expert on the various aspects of biogeochemical model parameterizations, but I understand the most important feedbacks between the different compartments of the BGC model and the importance of the physical forcing. In this work, there are some assumptions that can be considered as simplifications (e.g., 1D model, physical forcing, length of simulations etc.), but in my opinion there are all justifiable and there are other novelties that compensate for the study approximations.

Overall, I find the ms. worthy of publication in Biogeosciences after minor revisions. Please find below a list of comments that I would like the authors to address.

Specific comments:

1) P6, L183 and L188. Vertical eddy diffusivity is varying with depth or is set constant? Please clarify.

**Vertical eddy diffusivity varies with depth.  We now state:**

**"We simulate vertical mixing as a simple diffusive process using vertical eddy diffusivity coefficients that vary with depth and are estimated for each Lagrangian experiment using Thorpe-scale analyses from field measurements (Gargett and Garner, 2008)"**

2) P6, L196. Which model variables, in addition to the euphotic zone, you could have simulated? Please clarify why those variables were excluded from the simulation (e.g., computational cost?) and explain how this may affect model uncertainty in relation to other error assumptions.

**This question is not entirely clear to us.  We did not exclude any variables from the model.  We simulated all model variables in the depth range from the surface to the base of the euphotic zone as defined by the 0.1% light level (which was typically at depths ranging from 40 – 100 m and hence included approximately 10 – 20 model layers).   We made the decision to only model from the surface to the base of the euphotic zone (rather than from the surface down to an arbtitrary deeper depth, such as 1000 m) for two reasons: First, it does substantially decrease computational cost.  Second, the vast majority of our field measurements were made in the euphotic zone.  Thus, simulating the twilight zone would not give us substantial improvements in model fit (due to a lack of validation data below the euphotic zone), and it also would have added problems associated with defining boundary conditions (i.e., determining what the concentrations of each state variable should be at 1000 m depth would not have been feasible since most of them were not measured beneath the euphotic zone).**

3) P7, L237 and P8, L252-264. In the context of data assimilation, observational errors are often considered as a combination of instrument and representativity errors, the latter usually being the most important of all. The authors here quantify observational errors as the standard deviation of their measurements and/or the instrument error; if I understood correctly, representativity errors are not considered here. Are these errors relevant in terms of magnitude with observation representativity errors?

**We completely agree with the reviewer that representativity errors are often the dominant source of error.  However, we believe that the standard deviation of multiple distinct measurements inherently accounts for the most significant form of representativity error in our analyses.  Following Janjic et al. (2017) we consider three types of representatitivity error: I) error due to unresolved scales and processes, II) observation-operator error or forward model error, and III) errors associated with pre-processing or quality-control.  We note that since our data is mostly derived from direct *in situ* measurements, II and III are much less significant than they tend to be with, for instance remote sensing measurements.  We thus believe that error due to unresolved scales and processes is the dominant component in representativity error for our study. These unresolved scales and processes include such phenomena as temporal variability in vertical mixing or surface irradiance (i.e., inaccuracy of our steady-state physical forcing), diel variability in phytoplankton carbon:chlorophyll ratios, internal waves that displace communities upwards or downwards, etc. When we state that we used the standard deviation of our measurements, these are measurements from different sampling points within a model layer during the Lagrangian experiment (i.e., different times and depths).  This variability from one measurement to another thus incorporates representativity error (or at least the portion of this due to unresolved scales and processes) along with measurement error.  Typically, this standard deviation (which incorporates representativity error + instrument error) is the error that we used.  However, in the rare cases where the standard deviation was less than expected instrument error (which can happen, for instance if four nitrate measurements all returned a value of 0.4 mmol m$^{-3}$), we used the instrument error.**

4) I am confused with the threshold limit "detlim" referred as "experimental detection limit". How this threshold is defined? I see that the "detlim" depends on indeces i,j,k and that k-index is not an option for the observations; why? I think the authors should provide more explanations regarding the "detlim" threshold, because the cost function decrease (after several iterations) largely depends on this (at least this is what I understand from the definitions of J(p) and error_i,j,k at the end of page 7).

**The experimental detection limit varies for each measurement type. For instance, for particulate nitrogen the observational detection limit was 0.2 mmol N $m^{-3}$. This means that when values are below this (i.e., a measurement of 0 mmol N $m^{-3}$, we have no knowledge of whether the actual value was 0.001, 0.01, or 0.1 mmol N $m^{-3}$). Thus we cannot penalize the model if it returns any value less than the detection limit when the observation is also less than the detection limit. So if, for instance, the observation was 0.1, but the model returned a result of 0.02 we cannot say that there is any model mismatch at all (since both are less than the detection limit). In practice, the actual value of detlim for each measurement was not very important to our results, because observations were seldom less than detlim. However, this formal definition is necessary with log-normally distributed errors, because occasionally the reported observation value was zero (or even negative, in the case of NPP) and since the model can never take on values less than or equal to zero, this would lead to an infinite cost.**

5) Overall, in the data assimilation Section 2.4, it is not clear to me which model variables consist the control vector e.g., is it the same with the model state vector described in Table 1 (or not)? Please clarify.

**Just to be clear, since terminology can vary across disciplines, we assume that "control vector" is used here to denote the adjustable variables or parameters that determine the model's predictions and hence model-data misfit. As such our control vector is the set of 102 model parameters that we allow to vary (given in Supp. Table 1). This is essentially all parameters *except* for TLIM (the temperature dependence of growth, grazing, and respiration rates), which we chose not to allow to vary because it is both fairly well constrained from measurements and because allowing it to vary would obfuscate interpretations of variability in other parameters. We note, however, that model results also depend on the initial conditions, boundary conditions (at the base of the euphotic zone), and physical forcing (temperature, vertical diffusivity, and surface irradiance), which we prescribe directly from field measurements and hence do not allow to vary.**

6) P18, L669-670 "our work shows that very different parameter sets can result in similar cost function values, despite generating distinctly different model outputs". This is an interesting result, but what does it means exactly (especially here where the cost function is different wrt variational approaches)? Please elaborate.

**Most medium- to high-complexity biogeochemical models still utilize an approach in which a model run with a single biogeochemical parameter set is used to reach their conclusions. Sometimes this parameter set is determined by**

manually "tuning" the model to approximately match a set of observations, while other times the parameter set is determined through formal data assimilation that seeks to find the parameter set that produces a global (or more frequently a local) minimum in a cost function relating model output to observations. Both of these approaches, however, seek to find a single "best" set of model parameters that can then be used for a model run, which will be used to interrogate aspects of the marine system (e.g., in our case to understand the different pathways of the biological carbon pump). Our study shows that in a high-dimensional system (as all medium- to high-complexity biogeochemical models are) distinctly different sets of parameters can match the observations equally well but produce very different model results. Indeed, all of the parameter sets identified by our $OEP_{MCMC}$ approach had approximately identical values of the cost function, but some produced model ecosystems in which mixing was the dominant pathway of vertical carbon transport while others produced ecosystems with sinking particles as the dominant pathway. With either a typical "tuning" procedure or a more formal variational data assimilation approach, investigators would arrive at a single parameter set that would predict either that the ecosystem was dominated by mixing or by sinking particles (or perhaps a 50/50 split) that would give them a false certainty about the behavior of the ecosystem. An ensemble approach, using different biogeochemical parameter sets, is necessary to diagnose this model uncertainty. We now explain this further:

"For instance, different sets of parameters (all with approximately equivalent mismatch to our extensive suite of field measurements) predicted distinctly different functioning of the BCP in the CCE coastal region (with some parameter sets suggesting that subduction is most important and others suggesting that sinking particles are most important, Fig. 9b) and in the Costa Rica Dome (where some parameter sets suggested sinking was responsible for almost all carbon export, compared to other parameter sets that suggested almost equal importance of active transport, Fig. 9d). The results of a typical variational-adjoint data-assimilation approach (or any approach that determines results from a single "best" parameter set) would have selected only one of these possible parameter sets and assumed that it accurately depicted the ecosystem; our results more accurately quantify this true uncertainty."

7) P19, L690-692 "A further study (Anugerahanti et al., 2020) simultaneously perturbed physical circulation fields and the biogeochemical model and found that results were most sensitive to variability in the biological model". Vervatis et al., (2021a) and (2021b) performed ensemble simulations, using a 3D high-resolution ocean physics and

biogeochemical coupled model, to investigate unresolved scales and processes, perturbing (1) only ocean physics, (2) only BGC sources and sinks, and (3) both physics and BGC simultaneously, and found that uncertainties in physical forcing and parameterizations have larger impact on chlorophyll spread (and other BGC variables) than uncertainties in ecosystem sources and sinks. Moreover, this had an impact on increment analysis correction and on empirical consistency between model-data misfits, using various datasets (e.g., SST, SLA, total CHL and/or class-based PFTs). I think part of this information would improve the quality of the paper. This is merely a suggestion and I leave it up to the authors to decide if it is relevant to their work.

**Thank you for pointing us to these recent studies. We have added their results to our discussion.**

Minor comments:

1) P1, L26. Please avoid acronyms in the abstract e.g., CCE.

**Thank you for noting this. We will correct in the revised version.**

2) P7, L250. Do you mean $N_{O,i,j}$ instead of $N_{M,i,j}$?

**Yes, thank you for catching this. We had originally used 'M' for measurement and changed to 'O' for observation, but clearly missed one spot.**

Best regards,

V. Vervatis

References:

Vervatis, D. V., P. De Mey-Frémaux, N. Ayoub, J. Karagiorgos, M. Ghantous, M. Kailas, C.-E. Testut and S. Sofianos, 2021: Assessment of a regional physical-biogeochemical stochastic ocean model. Part 1: Ensemble generation, Ocean Modelling, 160, 101781, https://doi.org/10.1016/j.ocemod.2021.101781.

Vervatis, D. V., P. De Mey-Frémaux, N. Ayoub, J. Karagiorgos, S. Ciavatta, R.J.W. Brewin and S. Sofianos, 2021: Assessment of a regional physical-biogeochemical stochastic ocean model. Part 2: Empirical consistency, Ocean Modelling, 160, 101770, https://doi.org/10.1016/j.ocemod.2021.101770.

**Anonymous Reviewer #2:**

Summary and recommendation: This study uses a 1-dimensional ecosystem model to assimilate data from Lagrangian experiments in the Costa Rica Dome, California Current, Southern Ocean (Chatham Rise) and Gulf of Mexico. The authors use a Monte Carlo approach to assess the uncertainty in model predictions, compare the model predictions to observations within each region, and assess the export mechanisms (gravitational, mixing, and migration) in each region within their model. They find that the gravitational pump is most important in most regions, followed by the mixing pump and then the migration pump. The manuscript is well written and the results are clearly presented for the most part, so I recommend publication subject to minor revisions to address the points below.

The main strength of this study is that it uses a wide variety of in-situ data (rates, biomass, chemical tracers etc.) from several different ecosystems, which allows the many (>100) parameters of their model to be reasonably constrained, and that they use a MCMC approach to quantify the uncertainty in their model predictions. One weakness of this study is that the model is 1-dimensional and neglects horizontal transport and connectivity, as well as only resolving the euphotic zone, but this weakness is thoroughly discussed by the authors. Another weakness that is not as well addressed is why the model was not used for predictions outside the assimilation regions. I was hoping that the authors could also provide results from their model for regions that were not assimilated into the model, i.e. to extrapolate to other regions so as to produce global maps of export by these different mechanisms, or at least maps of export ratio. Without such an extrapolation to larger time and space scales the study is interesting but lacks a prediction that can be compared to other export models (except in the 4 regions that provided data that was assimilated into the model, on short timescales). It is also odd that the authors fail to mention the data-assimilation model of DeVries and Weber (2017), given their relatively thorough review of other assimilation models in the introduction and elsewhere, as well as the recent study by Nowicki et al (2022) with a quite similar title. Some other minor issues are noted below.

**The main reason that we do not extrapolate the model to other regions in its current form is that the simulations that we conducted require incredibly detailed and robust ecosystem analyses spanning physics, biogeochemistry, and ecology in order to prescribe accurate initial conditions, boundary conditions, and forcing.  There are very few programs that have sufficiently measured all of these processes simultaneously.  We thus believe that the appropriate way to extrapolate these results to other regions is to conduct fully-coupled four-dimensional ensemble simulations based on the parameter sets determined in this study.  That will require re-coding the model into a global circulation model**

**(e.g., HYCOM or MITgcm) and conducting hundreds of global simulations. We thus consider it beyond the scope of this manuscript but hope to address it in a future study.**

**Thank you for reminding us of the DeVries and Weber (2017) study. We had focused on data assimilation with fully mechanistic models, but we agree that the DeVries and Weber (2017) study, which uses satellite remote sensing to force a more simplified model of the biological pump, utilizes an interesting data assimilation approach that is certainly worth discussing in the context of our manuscript. We did not cite the Nowicki et al. (2022) study for the simple reason that it had not been published (and we hence were completely unaware of it) when we submitted this manuscript. We now cite these studies in multiple locations in the manuscript, as appropriate.**

- Lines 70-95: This discussion is missing the pioneering data assimilation models of Schlitzer (e.g. 2000; 2002) as well as the more recent work by DeVries and Weber (2017) and Nowicki et al. (2022).

**We thank you for pointing these out and now cite these articles in the revised manuscript.**

- Lines 136-145: Here two different configurations of the model are mentioned, one that only resolves the euphotic zone and one that resolves deeper layers that the zooplankton can migrate towards. This makes it sound like the model is run in both of these configurations, but then later (line 197) they say that only the euphotic zone configuration was used. So, I recommend to remove discussion of the other configuration to avoid confusion.

**This manuscript functions, in part, as a description of a new model system. We thus believe it is appropriate to describe both configurations, even though we only actually use one. However, in the revised draft, we make it more clear that we only used the euphotic zone-only configuration.**

- Figure 3: Some of the variables appear to have a peak probability that is at the limit of their allowable range. Does this represent a flaw in the model, or that the allowable

range should be widened in order to better capture the values of these parameters in the model simulations?

**We do not consider this a flaw. Instead, it demonstrates that the data is successfully constraining the possible solutions. For example, consider the respiration term for small phytoplankton (res$_{SP}$ shown in Fig. 3). This term represents the proportion of small phytoplankton biomass that is lost each day due to basal metabolism. This term is uncertain, because it is not trivial to separate out phytoplankton respiration from respiration of other microbes in field measurements. Hence, we assumed a priori that it could be anywhere between 0 and 0.1 d$^{-1}$ (with an initial guess of 0.002 d$^{-1}$). Model results showed that values of res$_{SP}$ greater than ~0.01 are inconsistent with the observations. This thus puts a strong constraint on that parameter. Similarly, consider the excretion parameter (exc$_{SP}$ also in Fig. 3), which quantifies the proportion of the gross primary production of small phytoplankton that is excreted (i.e., active metabolic processes). Based on prior results, we assumed that this parameter could take a value between 0 and 0.3 (with an initial guess of 0.135). However, model results showed that values of exc$_{SP}$ on the lower end of that range were inconsistent with the observations and that most likely exc$_{SP}$ is > 0.25). It is certainly possible to question whether or not our prior estimated ranges are the best choices (as with priors in Bayesian statistics there can always be critiques of what ranges should be allowable). However, we chose ranges that we believed were ecological and biologically realistic based on a combination of field and laboratory measurements and previous model results and parameterizations. We believe that the posterior distributions of parameters derived from our ensemble approach provide important information for more objective priors in future work (i.e., in future projects incorporating new datasets, we will likely model the prior of exc$_{SP}$ as a normal distribution with a mean of 0.27 and a standard deviation of 0.027 based on the results of the current manuscript's analysis.**

- Discussion of the mixing pump in general: For the mixing pump especially (more so than the other export pathways) it is important on what timescale the material remains exported, and can therefore contribute to carbon sequestration. Since the authors are running short timescales experiments (30 days) they should clarify that their modeled export is over that time horizon, and would not necessarily be the same as export over the course of the year. It should also be mentioned that the large-scale physical mixing pump (e.g. Ekman pumping) is not captured. The authors should speculate as to whether their model would provide an over- or underestimate- of the mixing pump

export on timescales relevant to carbon sequestration (> 1 year). This discussion could augment what the authors already have in lines 622-631.

**The reviewer is certainly correct that our model gives little information about the long-term fate of carbon leaving the euphotic zone via the mixing pump. In fact, it gives little information about the long-term fate of carbon leaving the euphotic zone via any mechanism; the model as currently formulated specifically asks the question of how much carbon is leaving the euphotic zone and by which process without quantifying the depth at which any of that carbon is sequestered and hence its long-term carbon storage potential. We certainly expect that carbon sequestration temporal horizons will vary for each of the different export mechanisms (as an aside, we are looking at this in detail with datasets from the California Current Ecosystem using different approaches). We believe that answering questions about the length of time that carbon is sequestered will require three-dimensional coupled runs (which we plan to do in the future). We believe that we make this clear in lines 656-675:**

**"One aspect of the BCP that our current euphotic-zone only simulations do not address is sequestration efficiency in the mesopelagic (Kwon et al., 2009; Marsay et al., 2015; Buesseler and Boyd, 2009). It is reasonable to surmise that the remineralization length scale will vary for different BCP pathways and be regionally variable as well. With gravitational flux, typically ~50% of particles will sink 100 m beneath the euphotic zone before remineralization, although remineralization length scales are highly variable and the spatial patterns are poorly understood (Buesseler and Boyd, 2009; Marsay et al., 2015). Meanwhile, vertically-migrating zooplankton typically reside at depths of 200 – 600 m during the day and hence respire the majority of their carbon dioxide at this depth (Bianchi et al., 2013b), although it is unclear how the inclusion of mortality at depth into our understanding of active transport will affect the overall depth of penetration of actively transported carbon into the deep ocean. Stukel et al. (2018b), suggested that subducted particles in the southern CCE are mostly remineralized near the base of the euphotic zone with little penetration into the mesopelagic, although in regions with deep convective mixing, signatures of subduction show substantial transport into the deep ocean (Omand et al., 2015; Llort et al., 2018). Nowicki et al. (2022) estimated that gravitational flux and active transport have similar sequestration time scales, but that sequestration times for mixing were much shorter. In contrast, Boyd et al. (2019) surmised that active transport may have the greatest sequestration efficiency, followed by vertical mixing, then gravitational flux, although their synthesis was only able to draw from the few studies that have quantified these processes and they note that determining the sensitivities of sequestration efficiencies to environmental**

**drivers is crucial to predicting climate change impacts on marine carbon sequestration. We believe that future incorporation of our model ensemble approach into three-dimensional coupled modeling frameworks could be an important step forward in understanding the magnitude, and uncertainty in these processes."**

**With respect to which aspects of the mixing pump are included, it is a little bit complex, because while we model a one-dimensional water column using only diffusive processes, the eddy diffusivity coefficient is based on Thorpe-scale analysis, which utilizes the magnitude and frequency of density instabilities to estimate shear-generated mixing. Thorpe-scale analyses thus do not explicitly map onto any of the different mechanisms of the mixing pump (as defined by Body et al. 2019 or Levy et al. 2013) but can be impacted by all of them. As with quantification of carbon sequestration timelines, we believe that three-dimensional coupled modeling is necessary to explicitly look at the different aspects of the mixing pump. We are not comfortable speculating as to whether the current approach over or underestimates the magnitude of the mixing pump, although we do note that the results derived for the CCE in this manuscript were not too dissimilar from results of an entirely independent approach (three-dimensional Lagrangian particle tracking, Stukel et al. 2018). We do, however, try to make these distinctions about the different mechanisms of the mixing pump (and the uncertainty associated with estimating the mixing pump in a one-dimensional framework) clearer in the revised manuscript by stating: "Our vertical mixing results should be considered with some caution due to our overly simplified one-dimensional physical framework, which conflates distinct processes including mesoscale subduction, diapycnal diffusing, mixed layer entrainment and detrainment, and gyre-scale Ekman pumping" We also state that: "More realistic estimates for all regions could be derived by coupling NEMUROBCP and our parameter ensembles to a three-dimensional ocean simulation."**

- Figure 11: From this figure it is hard to assess how the model-predicted and observed export compare. It would be good to show a scatterplot of the correlation between modeled and observed export in one figure, in addition to what is shown here.

**Our goal with this figure was not to show model data matchups. For an explicit comparison of how well the model sinking particle flux matched sediment trap data, we included Fig. 6. The purpose of Fig. 11 is to show that the model accurately estimates a realistic spread in the relationship between gravitational**

**flux and net primary productivity (e.g., both the model and the data agree that across a wide range of NPP export efficiency typically ranges between ~3% and 50%, without a strong correlation between export and efficiency). Since the reviewer is curious, we have created a scatterplot of model vs. observations (below), which can be included with the manuscript if requested. However, we believe that Fig. 6 conveys more information.**

[Figure]

- Several times throughout the paper the acronym SalpPOOP is mentioned, but never defined. I assume this is the Southern Ocean experiment that is elsewhere referred to as Chatham Rise??

**Yes, you are correct. Thank you for noting that we forgot to define it. We have defined in the revised draft. The cruise acronym is Salp Particle expOrt and Oceanic Production**

- Line 643 ff: The study of Nowicki et al (2022) assessed the sequestration times of the different export pathways and is highly relevant to this discussion.

**As mentioned before, the Nowicki et al. (2022) study was not available when we submitted the manuscript. It is, of course, highly relevant and we have incorporated its insights throughout the introduction and discussion.**

- Section 4.2: Again this discussion is oddly missing reference to the data assimilation studies of DeVries and Weber (2017) and Nowicki et al (2022)

**Agreed.  Our failure to include the DeVries and Weber study was a definite oversight.  The Nowicki study was simply not included because it had not been published when we submitted the manuscript.  Both are now cited.**

References:

Schlitzer, R. (2000). Applying the adjoint method for biogeochemical modeling: export of particulate organic matter in the world ocean. Geophysical Monograph-American Geophysical Union, 114, 107-124.

Schlitzer, R. (2002). Carbon export fluxes in the Southern Ocean: results from inverse modeling and comparison with satellite-based estimates. Deep Sea Research Part II: Topical Studies in Oceanography, 49(9-10), 1623-1644.

DeVries, T., & Weber, T. (2017). The export and fate of organic matter in the ocean: New constraints from combining satellite and oceanographic tracer observations. Global Biogeochemical Cycles, 31(3), 535-555.

Nowicki, M., DeVries, T., & Siegel, D. A. (2022). Quantifying the carbon export and sequestration pathways of the ocean's biological carbon pump. Global Biogeochemical Cycles, 36(3), e2021GB007083.

---

## Author Response (AR2)

**Associate Editor decision: Publish subject to minor revisions (review by editor)**
by Marilaure Grégoire
**Comments to the author**:
Dear Michaël Stukel and co-authors,

I have read the answers you provided to the comments of the two reviewers. I am satisfied with your answers but I found that, in some instances, the manuscript has not been modified to clarify reviewers' questions.

**Thank you for this opportunity to revise our manuscript. We have made the suggested revisions as outlined below.**

Also, before acceptance of your work I would like that you modify the manuscript to answer the following comments:

1) comment of reviewer #1 on the representativity error,

**At lines 278 – 285 we have added the text:**

**"We note that observational uncertainty can result from both instrument error and representativity error, and while we explicitly incorporate instrument error, we do not directly include all sources of representativity error. Representativity error refers to error due to unresolved scales and processes, observation-operator error, and errors associated with pre-processing and quality control (Janjić et al., 2018). Since our data is derived from direct in situ measurements, the latter two sources of representativity error are likely much less significant than errors resulting from unresolved scales and processes. Because we incorporate the standard deviation of multiple measurements taken at different depths and sampling times within a model layer in our measurement uncertainty, we include this dominant source of representativity error."**

2) comment of reviewer #1 on the detection limit,

**At lines 264 – 267 we have added the following text:**

**"Detection limits varied depending on measurement type. In practice the actual value of $detlim_{i,j,k}$ was not very important to our results, because observations were seldom less than $detlim_{i,j,k}$. However, this formal definition is necessary with log-normally distributed errors, because occasionally the reported observational value was zero (or even negative)."**

3) comment of reviewer #2 about the distribution of variables that have a probability peak at the limit of their allowable range.

**At lines 348 – 353 we have added the text:**

**"We note that some well-constrained parameters were constrained by the data to fall within narrow bands near the middle of their prior allowable range (e.g., $V_{max,SP}$, Fig. 3) and others were constrained to the edges of their allowable ranges (e.g., $\alpha_{SP}$, Fig. 3). While the latter case shows sensitivity of our model to our chosen priors, we do not consider this a flaw. Instead, it demonstrates that the data is providing strong constraint on the possible values of these parameters and effectively providing guidance for constraining these parameters in future studies."**

Many thanks for your efforts,

Kind regards,

Marilaure Grégoire, Associate editor.

---

## Author Response (AR3)

As requested, we have removed the colors from Table 1.

-Thanks,

 Mike